# The luminal ring protein C2CD3 acts as a radial in-to-out organizer of the distal centriole and appendages

Eloïse Bertiaux[1], Vincent Louvel[1], Caitlyn L. McCafferty[2], Hugo van den Hoek[2], Umut Batman[1], Souradip Mukherjee[1], Lorène Bournonville[1], Olivier Mercey[1], Isabelle Méan[1], Ricardo D. Righetto[2], Adrian Müller[2], Philippe Van der Stappen[2,3], Garrison Buss[4], Jean Daraspe[5], Christel Genoud[5,6], Tim Stearns[4¤], Benjamin D. Engel[2], Virginie Hamel ©[1]*, Paul Guichard ©[1]*

**1** Department of Molecular and Cellular Biology, Sciences III, University of Geneva, Geneva, Switzerland, **2** Biozentrum, University of Basel, Basel, Switzerland, **3** Swiss Nanoscience Institute, University of Basel, Basel, Switzerland, **4** Department of Biology, Stanford University, Stanford, California, United States of America, **5** Electron Microscopy Facility, Biophore, University of Lausanne, Lausanne, Switzerland, **6** School of Life Sciences, Ecole Polytechnique Fédérale de Lausanne, Lausanne, Switzerland

¤ Current address: The Rockefeller University, New York, New York, United States of America.
* virginie.hamel@unige.ch (VH); paul.guichard@unige.ch (PG)

## Abstract

Centrioles are polarized microtubule-based structures with appendages at their distal end that are essential for cilia formation and function. The protein C2CD3 is critical for distal appendage assembly, with mutations linked to orofaciodigital syndrome and other ciliopathies. However, its precise molecular role in appendage recruitment remains unclear. Using ultrastructure expansion microscopy (U-ExM) and iterative U-ExM on human cells, together with in situ cryo-electron tomography (cryo-ET) on mouse tissues, we reveal that C2CD3 adopts a radially symmetric 9-fold organization within the centriole's distal lumen. We show that the C-terminal region of C2CD3 localizes close to a ~100 nm luminal ring structure consisting of ~27 nodes, while its N-terminal region localizes close to a hook-like structure that attaches to the A-microtubule as it extends from the centriole interior to exterior. This hook structure is adjacent to the DISCO complex (MNR/CEP90/OFD1), which marks future appendage sites. C2CD3 depletion disrupts not only the recruitment of the DISCO complex via direct interaction with MNR but also destabilizes the luminal ring network composed of C2CD3/SFI1/centrin-2/CEP135/NA14, as well as the distal microtubule tip protein CEP162. This reveals an intricate "in-to-out" molecular hub connecting the centriolar lumen, distal microtubule cap, and appendages. Although C2CD3 loss results in shorter centrioles and appendage defects, key structural elements remain intact, permitting continued centriole duplication. We propose that C2CD3 forms the luminal ring structure and extends radially to the space between triplet microtubules,

**Data availability statement:** Reconstructed cryo-electron tomography (cryo-ET) datasets of photoreceptor and mouse tracheal epithelial cells have been deposited in the Electron Microscopy Data Bank (EMDB) under accession codes EMD-55390 and EMD-55391 (photoreceptor cells) and EMD-55386 (mouse tracheal cells). The corresponding sub-tomogram averages from mouse tracheal cells are available under accession codes EMD-55393 (hook) and EMD-55394 (distal ring). Raw cryo-ET data of mouse tracheal cells are deposited at the Electron Microscopy Public Image Archive under accession code EMPIAR-13055. U-ExM images of all figures has been deposited to the Yareta, the archive portal of the University of Geneva: https://doi.org/10.26037/yareta:gmr7hvps5vbhlmeloxnoqktl5u.

**Funding:** This work was supported by an EMBO Long-Term Fellowship (no. 284–2019) awarded to E.B. https://www.embo.org/funding/fellowships-grants-and-career-support/postdoctoral-fellowships/; a SNSF Postdoctoral Fellowship TMPFP3_224900 awarded to C.L.M. https://www.snf.ch/en/m1NtWp4nTELQixlu/funding/horizon-europe-swiss-postdoctoral-fellowships; SNSF Project Grant (grant PP00P3_187198 & 310030_205087) awarded to P.G. and V.H.; https://www.snf.ch/en/WAvYcY7awAUGolST/funding/projects/projects-in-all-disciplines the Pro Visu and Gelbert Foundations supporting V.H. https://www.provisu.ch/ https://fondation-gelbert.ch/; ERC Consolidator Grant ISAC, administered in Switzerland by SERI (contract MB22.00075) awarded to P.G.; https://www.sbfi.admin.ch/sbfi/en/home.html; bridging postdoctoral funds provided by the Biozentrum to B.D.E.; https://www.unibas.ch/en.html; NIH grant R35GM130286 awarded to T.S https://www.nih.gov/ The funders/sponsors had no role in study design, data collection and analysis, decision to publish, or preparation of the manuscript.

**Competing interests:** The authors have declared that no competing interests exist.

**Abbreviations:** ALI, air–liquid interface; cryo-ET, cryo-electron tomography; EM, electron microscopy; FA, formaldehyde; iU-ExM, iterative ultrastructure expansion microscopy; LDR, luminal distal ring; MTECs, mouse tracheal epithelial cells; OFD, oral-facial-digital; ROI, region of interest; SA, sodium acrylate; U-ExM, ultrastructure expansion microscopy.

functioning as an architectural hub that scaffolds the distal end of the centriole, orchestrating its assembly and directing appendage formation.

## Introduction

Centrioles are cytoskeletal organelles with important roles in cell division, cell polarity, and serving as basal bodies for the formation of cilia/flagella. These organelles are 9-fold microtubule-based cylinders with a diameter of ~200 nm and a polarized composition of microtubule-associated proteins along their proximal to distal axis. The proximal region contains the cartwheel structure, essential for templating the formation of a new procentriole, dictating the 9-fold symmetry, and stabilizing centriole intermediates [1–5]. The central region features an inner scaffold lining the internal wall of the centriolar microtubules, which plays a role in maintaining centriole architecture integrity [6,7]. At the distal end, mature centrioles are decorated on the outside of the microtubule wall with sub-distal and distal appendages, which mediate centriole docking to the plasma membrane and ciliogenesis [8–10]. Each of the nine distal appendages are connected to a doublet or triplet microtubule and consist of numerous proteins organized in layers, forming a large fibrous structure that gives the distal centriole a diameter of ~600 nm when measured from the tips of the appendages [11]. Core components of the distal appendages include CEP83, CEP89, SCLT1, CEP164, and FBF1, which are recruited to the distal ends of mature centrioles and assembled during the G2/M phase [12]. The appendages are anchored via a complex composed of CEP90, MNR, OFD1, and FOPNL. In this complex, also called the DISCO complex, MNR recruits OFD1, FOPNL, and CEP90, which in turn recruit CEP83, CEP89, and CEP164 [10,13]. Unlike appendages, which are only present on mature centrioles, this anchoring complex is present from the very beginning of procentriole assembly [13].

The mechanism underlying appendage assembly and the recruitment of these complexes is incompletely understood. However, C2CD3 appears to be crucial for this process and has been proposed to be an upstream regulator [14–16]. It has been demonstrated that C2CD3 co-localizes and physically associates with OFD1 at the distal ends of centrioles [16], suggesting that C2CD3 is involved in recruiting the anchoring complex. However, it was recently observed that C2CD3 does not localize outside the centriolar barrel on appendages or at the level of the anchoring complex but rather is located inside the centriole lumen. Using super-resolution microscopy techniques such as ultrastructure expansion microscopy (U-ExM), C2CD3 appears as a ring-like structure forming the luminal distal ring (LDR) complex together with other proteins such as Centrin, CEP135, and SFI1 [17]. High-resolution imaging combining U-ExM and dSTORM, an approach called Ex-STORM [18,19], identified a 9-fold symmetrical arrangement of C2CD3 proteins positioned near the A-tubule, leading to the proposal that distal appendage develop through a linear organization from inside to outside of C2CD3, microtubule triplets, MNR, CEP90, OFD1, CEP83, CEP89, SCLT1, and CEP164. While its internal localization suggests a potential

signaling or scaffolding role within the centriole lumen, the precise architectural mechanism by which C2CD3 establishes the 9-fold symmetry of distal appendage assembly remains unclear.

Electron microscopy (EM) studies of various ciliated and non-ciliated mammalian cells have occasionally reported a ring structure in the lumen of the distal centriole that could correspond to the ring-like organization of C2CD3. First described by Anderson in 1972 [20] in oviduct epithelium and later seen in pig kidney [21], and human HeLa cells [22,23], the ring structure is composed of 27 evenly-spaced nodes, resembling a "pearl necklace" of balls on a circular string. However, the molecular identity and function of this luminal ring are unknown.

Here, we combined U-ExM and cryo-ET to investigate the structure and molecular organization of the distal centriole. Using U-ExM, we mapped three regions of C2CD3 and found it exhibits nine-fold radial symmetry with an "in-to-out" architecture, spanning from the C-terminus in the lumen to the N-terminus between microtubule blades. In situ cryo-ET resolved that the luminal ring structure coincides with the localization of the C2CD3 C-terminus and the LDR but also revealed a "hook" connector between microtubule blades near the C2CD3 N-terminus. We show that C2CD3 interacts with appendage-anchoring proteins MNR and CEP90, and contributes to a broader distal complex including SFI1, Centrin-2, CEP135, and NA14/SSNA1. The latter protein co-localizes with the C2CD3 C-terminus and also displays 9-fold symmetry, suggesting its role in the LDR. Depletion of C2CD3 leads to short centrioles with abnormal CEP97 and CEP162 localization but retaining core structural features of the proximal and middle centriole regions, underscoring C2CD3's role in both distal architecture and microtubule cap composition.

Our study underscores the key architectural role of C2CD3 in formation of the appendages, the centriolar microtubule cap, and the LDR complex, suggesting that its structural organization from the lumen outward (i.e., "in-to-out") is key to coordinating the molecular architecture of the distal region of the centriole.

## Results

### Radial distribution of C2CD3 extending from centriole lumen to microtubule blade interspace

The human C2CD3 protein comprises 2,353 amino acids, with six C2-domains described as $Ca^{2+}$-dependent membrane-targeting modules (Fig 1A). The localization of C2CD3 was recently examined using super-resolution microscopy, revealing a ring structure with a diameter of 80–90 nm in U-ExM, and a 146.4 nm ring with nine distinct puncta when U-ExM was combined with STORM [17,18,24]. The discrepancy of these values may be due to differences in imaging modality (U-ExM alone or coupled with STORM), but it seems more likely that this reflects different positions of the recognized epitopes. Indeed, the smaller diameter was seen using an antibody detecting the C-terminus of C2CD3 whereas the larger diameter was seen using an antibody detecting the middle part of C2CD3 (Fig 1A). This suggests that this long protein of 2,353 amino acids adopts an extended conformation. To test this hypothesis, we used three antibodies recognizing C2CD3 at distinct regions across its length, one in the N-terminal portion of C2CD3 (N-term), one in the middle, and one recognizing the C-terminal part of the protein (C-term) (Fig 1B–1D). We found that while the antibody recognizing C2CD3's C-term yielded an 74 ± 13 nm ring signal localization as previously shown (Fig 1D and 1E), the middle region revealed a ring with a diameter of 122 ± 13 nm (Fig 1C and 1E) and the N-term antibody gave a ring with the largest diameter of 138 ± 13 nm (Fig 1B and 1E), closest to the distal tubulin signal of 165 ± 13 nm (Fig 1E). This suggests that the C2CD3 protein extends from the internal lumen of the distal centriole region to the microtubule wall. Interestingly, we noticed that on the longitudinal axis, the C2CD3 C-term appeared more proximal than the middle and N-term of the protein, suggesting that the conformation is not totally flat but slightly inclined along the long-axis of the centriole barrel (Fig 1F and 1N).

To further refine our understanding of the C2CD3 organization and test whether the three mapped domains of C2CD3 display a 9-fold symmetry (as previously shown for the middle part by coupling U-ExM and STORM [18]), we turned to iterative ultrastructure expansion microscopy (iU-ExM), which enables a 16-fold expansion of the biological specimen with spatial resolution of about 10–20 nm [25]. Using this approach, we revealed a 9-fold symmetrical localization pattern of

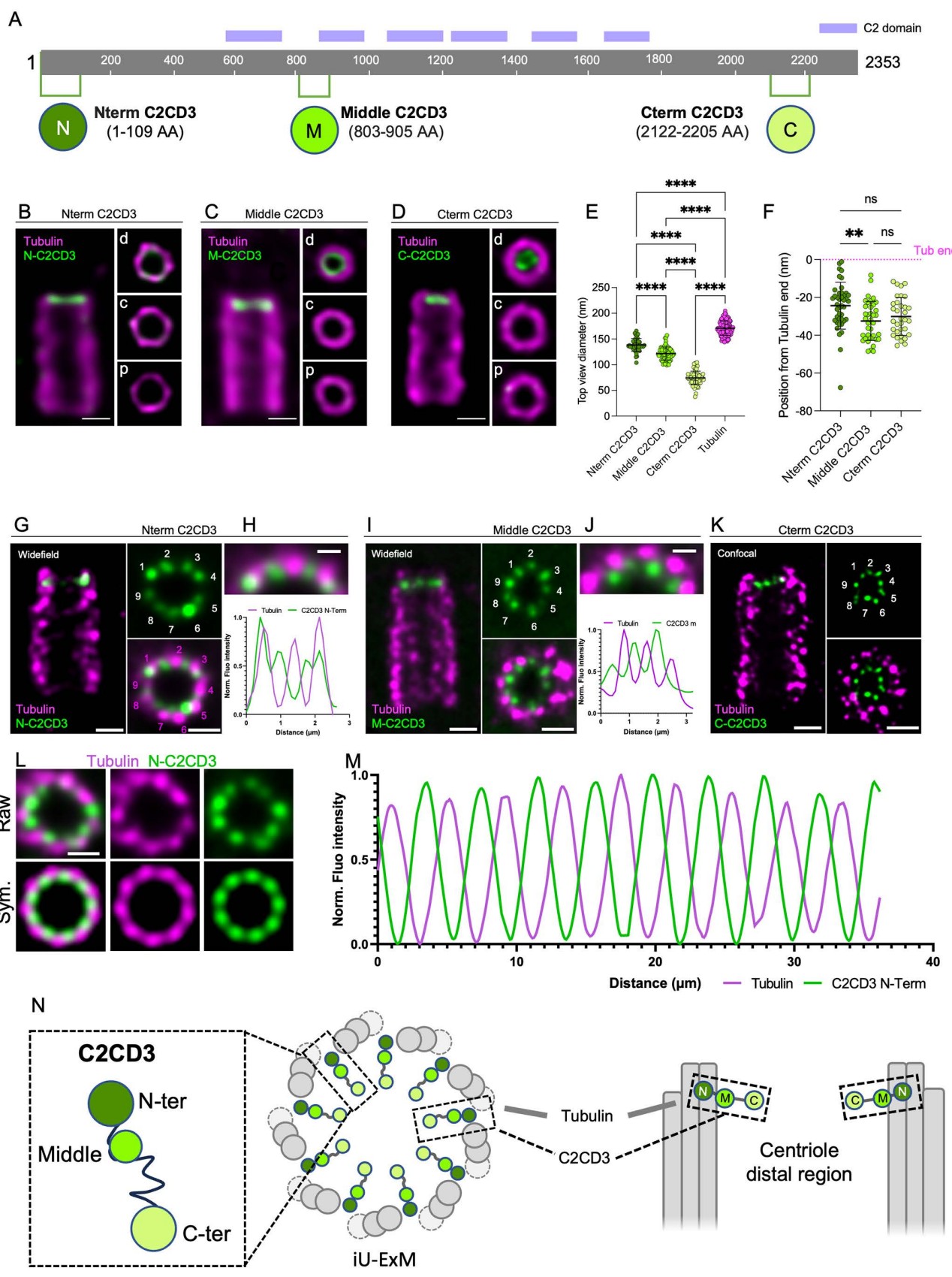

**Fig 1. C2CD3 exhibits nine-fold radial symmetry and distal localization in centrioles. (A)** Schematic of the C2CD3 protein showing its domain architecture and the epitopes recognized by three antibodies targeting the N-terminal, middle, and C-terminal regions. **(B–D)** Confocal images of expanded U2OS centrioles stained for α/β-tubulin (magenta) and C2CD3 (green) using antibodies against the N-terminal (B), middle (C), and C-terminal (D) epitopes. Right panels: top-down views along the centriole axis, showing C2CD3 localization within the distal region (d, distal; c, core; p, proximal). Scale bars: 100 nm. **(E)** Normalized diameters of C2CD3 signals relative to tubulin. Mean ± SD: tubulin, 165 ± 13 nm ($n = 177$); N-terminal, 138 ± 14 nm ($n = 11$); middle, 122 ± 13 nm ($n = 17$); C-terminal, 74 ± 13 nm ($n = 17$). Two-way ANOVA: ***$P < 0.0001$ for all pairwise comparisons (N-term vs. middle, N-term vs. C-term, middle vs. C-term). Data from three independent experiments. **(F)** Quantification of the axial positioning of C2CD3 relative to the distal tubulin signal (dotted magenta line). Mean ± SD: N-terminal, −24 ± 12 nm ($n = 42$); middle, −33 ± 10 nm ($n = 37$); C-terminal, −30 ± 10 nm ($n = 33$). Data from three independent experiments. Statistical analysis: two-way ANOVA (N-term vs. middle, $P = 0.0082$; N-term vs. C-term, ns $P = 0.1376$; middle vs. C-term, ns $P = 0.5671$). **(G, I, K)** Widefield (G, I) and confocal (K) images of iteratively expanded human centrioles stained for tubulin (magenta) and C2CD3 (green), using N-terminal (G), middle (I), or C-terminal (K) antibodies. Side and top views are shown. Scale bars: 100 nm. **(H, J, K)** Top panels: cropped top-view images from (G) and (I). Bottom panels: line profiles of tubulin and C2CD3 intensity (N-terminal in H; middle in J), showing C2CD3 localization between microtubule triplets. **(L)** Symmetrized top view of N-terminal C2CD3 distribution in iU-ExM. Scale bar: 100 nm. **(M)** Intensity profile of tubulin and N-terminal C2CD3 after symmetrization. **(N)** Model illustrating the spatial organization of C2CD3 within the distal lumen of the centriole. The detailed statistics of all the graphs shown in the figure are included in the S1 Data file.

C2CD3 for each of the antibodies covering the three distinct regions: N-term, middle, and C-term (Figs 1G–1L, S1A, and S1B). We also analyzed C2CD3's position relative to successive microtubule blades by imaging top views of centrioles for the N-term and middle region antibodies (Fig 1H–1J). We found that both the N-term and middle signals were aligned along a radial axis extending from the centriole center to the space between microtubule blades. While this observation had previously been made for the middle region by coupling U-ExM and STORM, it had never been reported for the N-term. To enhance the signal for this analysis, we applied 9-fold symmetrization to our images, revealing that the N-term is positioned close to the gap between microtubule blades but slightly more luminal than the peak tubulin signal (Figs 1L–1N, S1C, and S1D). Altogether, these results suggest a 9-fold radially symmetric organization of C2CD3, with the C-term pointing towards the centriole lumen and the N-term reaching towards the inter-microtubular space (Fig 1N).

### *In situ* cryo-ET reveals luminal ring and microtubule-associated hook structures that correlate with the position of C2CD3

Cellular EM and cryo-electron tomography (cryo-ET) have been instrumental in mapping numerous structural elements of the centriole. Several classical EM studies of resin-embedded cells have noted the presence of a ring structure (first described by Anderson, 1972 as the "Apical Ring") inside the lumen of the distal centriole [20,21]. While this structure has not been reported in isolated centrioles using cryo-ET, two recent target-guided in situ cryo-ET studies identified a ring structure composed of 27 evenly distributed density nodes [23,22]. However, the precise positioning and molecular composition of this ring have not been described.

To investigate whether C2CD3 associates with a specific structure at the distal centriole, we performed in situ cryo-ET on primary cilia from mouse photoreceptors (Fig 2). This cellular model offers the advantage that primary cilia are uniformly oriented, parallel to the axis of the retinal tissue, facilitating precise targeting of the centriole region just below the connecting cilium [26]. Importantly, by performing U-ExM on mouse retina, we confirmed that C2CD3 localizes to the distal lumen of centrioles, just below the transition zone and the connecting cilium (Fig 2A). Using cryo-FIB lift-out, we obtained cross-sections below the level of the connecting cilium, allowing us to visualize the entire distal region of the centriole, characterized by the termination of the C-microtubule in the triplet microtubule and the absence of surrounding membrane that marks the start of the connecting cilium (Fig 2B). In this tomogram, as well as in a separate tomogram of a thicker region containing an almost complete centriole in top view (S2 Fig), we observed a ring structure like that previously identified by cryo-ET of centrioles in cycling HeLa cells [22,27,23]. This ring appears to consist of one layer, positioned approximately 30–60 nm from the tips of the distal microtubules (S2 Fig, panel 1–3), coinciding with the localization of C2CD3 (Fig 1F).

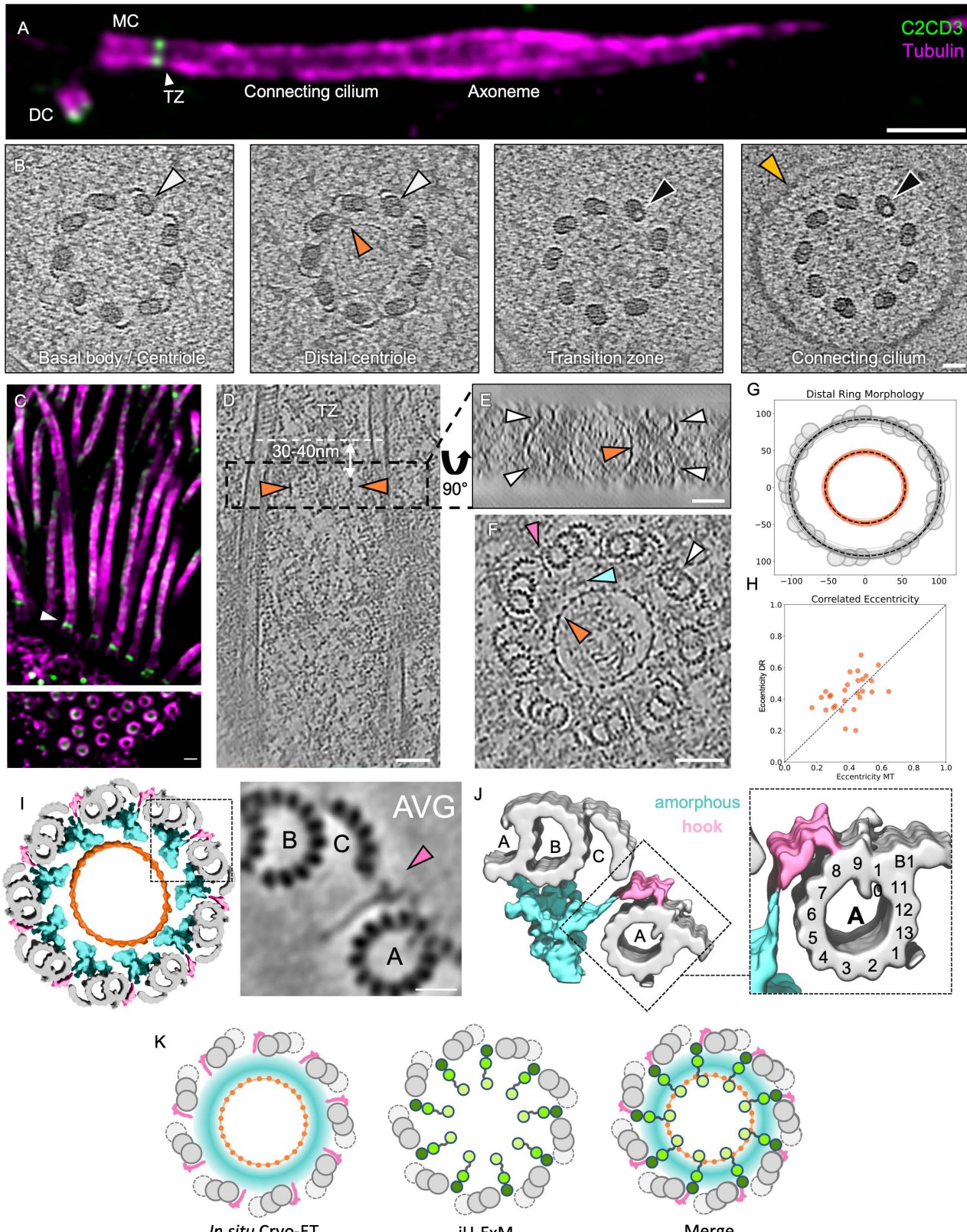

**Fig 2. C2CD3 localizes to a ring structure observed in the lumen of the distal centriole by in situ cryo-electron tomography. (A)** Confocal image of an expanded mouse photoreceptor cell immunolabeled for tubulin (magenta) and C2CD3 (green). DC, daughter centriole; BB, basal body; TZ,

transition zone; CC, connecting cilium. Scale bar: 500 nm. **(B)** In situ cryo-tomogram slices taken along the longitudinal axis of the centriole and connecting cilium. White arrowheads indicate microtubule triplets, black arrowheads indicate microtubule doublets, orange arrowheads mark the luminal ring structure, and yellow highlights the membrane. Scale bar: 50 nm. **(C)** Confocal image of an expanded mouse tracheal epithelial cell (MTEC) labeled for tubulin (magenta) and C2CD3 (middle epitope, green). Scale bar: 200 nm. **(D)** Longitudinal cryo-tomogram slice of a basal body from a mouse tracheal cell. Orange arrowheads mark the edges of the luminal ring structure, seen in cross-section. Scale bar: 50 nm. **(E)** 90° rotated top view of the boxed region in (D), showing the ring structure (orange arrowhead) encircled by the microtubule triplets (white arrowheads). Scale bar: 50 nm. **(F)** Top-view cryo-tomogram slice of the distal end of a basal body, highlighting the microtubule triplets (white arrowhead), ring structure (orange arrowhead), amorphous density between the ring and microtubules (blue arrowhead), and a distinct hook-like structure between the microtubule blades (pink arrowhead). Scale bar: 50 nm. **(G)** Cross-section analysis of the centriole and ring diameter. Individual ring structures are plotted in orange, microtubule walls are plotted in gray. Average diameters are plotted in dashed black lines. **(H)** Eccentricity of the distal ring (DR) structure is correlated with eccentricity of the surrounding microtubule (MT) wall. **(I)** 3D reconstruction of the distal centriole using subtomogram averaging of the hook-like structure region (S3D Fig) and the ring structure region (S3E Fig). Hook structure is shown in pink; microtubule triplets in gray; surrounding amorphous density in cyan. **(J)** 3D surface rendering of the subtomogram average from (I), showing the hook interacting between protofilaments A8 and A9 of the A-microtubule. **(K)** Proposed model integrating in situ cryo-ET and iU-ExM data, suggesting that C2CD3 is a structural component of the distal ring and radially extending to hook-like densities. The detailed statistics of all the graphs shown in the figure are included in the S1 Data file.

To gain further insights into distal centriole architecture, we next used in situ cryo-ET to image mouse tracheal epithelial cells (MTEC), as this cell type provides a high number of centrioles, enabling the acquisition of diverse cross-sectional views (top and side). After confirming that C2CD3 also localizes at the distal end of the basal bodies in mouse trachea (Fig 2C), we obtained several in situ cryo-tomograms passing through the distal centriole region as well as longitudinal views of the entire centriole (Fig 2D). Consistent with our observation in mouse retina, the distal region of mouse tracheal centrioles also contained a luminal ring structure, which resembles a "pearl necklace" with 27 evenly spaced nodes, suggesting an arrangement of 3 nodes per each of the centriole's 9 asymmetric units (Fig 2F). This ring structure had an average diameter of ~102 nm (long axis: $108.4 \pm 4.7$ nm, short axis: $96.5 \pm 4.6$ nm, $N = 28$ rings), and was located about 30–40 nm from the end of the basal body/beginning of the TZ (Fig 2D and 2E), which also corresponds to the position of the C2CD3 C-terminus (Fig 1F). Notably, the centriole cross-section was not perfectly circular but slightly ovoid. This ovoid deformation was mirrored by the shape of the ring, with both structures exhibiting variation in the same direction. A correlation analysis revealed that the eccentricity of the ring closely follows that of the microtubule wall, suggesting a structural coupling between the ring and the surrounding microtubule scaffold (Fig 2G and 2H).

To precisely relate the diameter of the luminal ring structure measured by cryo-ET to the diameters of radial signal for C2CD3 domains observed via U-ExM, we then normalized the averaged measurements between the two imaging modalities using the microtubule wall as an internal reference. Given that the apparent microtubule diameter differs slightly between U-ExM (165 nm, based on tubulin signal) and cryo-ET (195 nm, from the center of the microtubule triplets/doublets), we applied a proportional scaling factor. Specifically, we used a cross-multiplication approach, treating the more precise cryo-ET tubulin diameter as a reference to recalculate the diameters measured by U-ExM for the different domains of C2CD3 (Fig 1E). When scaled accordingly (e.g., 74 nm × 195 nm/165 nm ≈ 87 nm for the C-term), the antibodies recognizing C2DC3's C-term, middle region, and N-term yield radial signal with diameters of ~87 nm, ~140 nm, and ~163 nm, respectively. This registration of cryo-ET and U-ExM measurements revealed that the diameter of the C-terminal region of C2CD3 (87 nm) is the closest to the diameter of the luminal ring structure (102 nm), a radius difference of just 7 nm, while the middle and N-terminal regions of the protein extend beyond the ring toward the centriole's microtubule wall. Based on this analysis, we propose that the luminal ring structure corresponds to the LDR complex previously described by U-ExM [17] and is likely constructed at least in part by a region of C2CD3 close to its C-term.

Although we did not observe a well-defined radial 9-fold organization in the space between the luminal ring structure and the microtubule triplets, instead noting an amorphous density (Figs 2F and S3), we identified "hook-like" connecting structures projecting from this amorphous region (pink arrowheads). These hook structures were positioned between each microtubule triplet, aligned with the plane of the luminal ring structure, but were also detected up to $29 \pm 8$ nm more distal than the ring (Figs 2F, 2I, 2J, and S3A–S3C). This spatial arrangement resembles the localization pattern of C2CD3, which has an N-terminal region that is slightly more distal than its C-terminal counterpart (Fig 1E). Subtomogram

averaging of this region revealed that the hook structures contact the A-microtubule between the A8 and A9 protofilaments (Figs 2I, 2J, and S3D). Taken together, we propose that the ring structure observed in the distal lumen of the centriole corresponds to the previously identified LDR complex and that it is partially composed of C2CD3, with the C-term comprising part of the ring structure and the N-term positioned close to the hook connector between the microtubule blades (Fig 2K).

## Molecular architecture of the centriole's distal end

At the distal end of the centriole, in addition to the distal appendages, three major molecular complexes have been identified: (1) the distal cap, positioned atop the microtubule blades and composed in part of CEP97, CP110, CEP162, and CPAP [17,28,29]; (2) the LDR, located ~40–50 nm below the distal end of the centriole and comprising C2CD3, CEP135, Centrins, and SFI1 [17,30], and (3) the DISCO complex, composed of CEP90, MNR, OFD1, and FOPNL [10,13]. This latter complex was proposed to localize at the same longitudinal height as C2CD3, based on the reference position of the distal appendage protein SCTL1 [18]. However, its precise spatial relationship to the tubulin wall and distal end of the centriole remain uncharacterized.

Here, we set out to map these three complexes relative to the molecular architecture of the distal centriole. We used SFI1 and C2CD3 to mark the LDR, CEP162, and CP110 for the distal cap, and CEP90, MNR, and OFD1 for the DISCO complex. Additionally, we identified a novel distal component, NA14 (also known as SSNA1/DIP13), which localizes to the distal lumen (Fig 3A). Our U-ExM analysis revealed that NA14 forms a ring in the lumen of the distal end, with a diameter of 88 ± 15 nm, positioned −30 ± 13 nm from to the microtubule ends (Figs 3A–3D and S4). When scaled by our cryo-ET measurements, the diameter of NA14 is ~104 nm, which almost exactly overlaps the distal ring structure we see in our tomograms (~102 nm diameter). Like C2CD3 and other LDR components [17], NA14 is recruited early during centriole biogenesis, initially associating with microtubules before becoming internalized beneath the distal microtubule ends as elongation progresses (S4A–S4G Fig). This supports NA14 as another component of the LDR.

We further investigated the relative positioning of the DISCO complex by analyzing its distribution with respect to the luminal ring proteins C2CD3, SFI1, and NA14, as well as the distal cap proteins CP110 and CEP162 (Figs 3A, 3C, and S4). We found that C2CD3 (Nterm −24 ± 12 nm, Middle −32 ± 10 nm, and Cterm −30 ± 10 nm), NA14 (−30 ± 13 nm), and SFI1 (−30 ± 13 nm) are positioned ~30 nm from the tubulin ends, whereas CEP162 and CP110 are more distally located at −4 ± 19 nm and +28 ± 12 nm, respectively. In contrast, CEP90, MNR, and OFD1 reside at −56 ± 17 nm, −37 ± 11 nm, and −48 ± 16 nm, respectively, very close the LDR position in the longitudinal axis but slightly more proximal. These relative positions were confirmed via co-staining of NA14 and CEP90, C2CD3 C-term and CEP90, NA14 and MNR, and C2CD3 C-term and MNR (Figs 3D and S4J).

Given the observed proximity between the LDR and DISCO complexes, we next asked whether CEP90 aligns with the radial organization of C2CD3 and whether it could correspond to the position of the connector and hook structures seen attached to the external surface of the A-microtubule in cryo-ET. To test this, we performed iU-ExM staining, labeling tubulin alongside the C2CD3 middle region and CEP90 (Fig 3E). Analysis of top-viewed centrioles revealed that CEP90 is not aligned with the axis extending from the centriole center to the C2CD3 signal but instead forms an angle of 158.3° (Fig 3F). To determine whether this angle correlates with microtubules A, B, or C within the triplet, we modeled three possible configurations (Figs 3G and S4K). The best fit indicated localization of CEP90 on the A-microtubule with an angle of 155° (Fig 3G), aligning with the radial position of the hook structure seen in cryo-ET (Fig 2I–2K). We note, however, that while CEP90, C2CD3 N-term, and the hook structure are localized at the A-microtubule, their positions are offset longitudinally. CEP90 is located more proximal relative to the luminal ring structure (marked by the C-terminal of C2CD3 and NA14), while the hook structure (C2CD3 N-terminus) is positioned at the level of the luminal ring and extends further distal (Figs 3C and S3A–S3C). Other components of the DISCO complex are situated slightly more distal than CEP90, with MNR nearly longitudinally aligned with the luminal ring structure (Fig 3C).

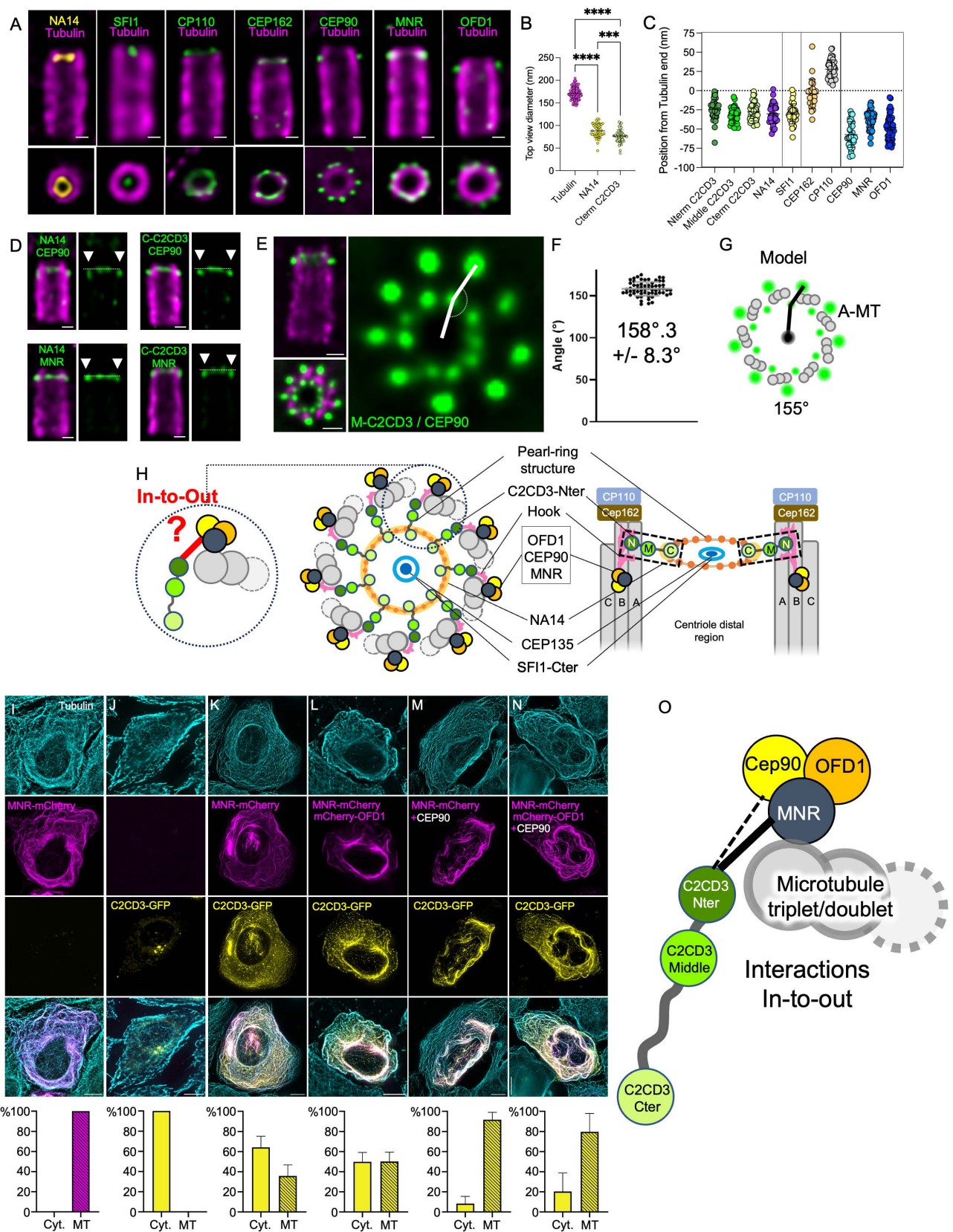

**Fig 3. C2CD3 forms a distal in-to-out connection that bridges the centriole lumen to the DISCO complex. (A)** Confocal images of expanded U2OS centrioles immunolabeled for tubulin (magenta), NA14 (yellow), and one of the following proteins (green): SFI1, CP110, CEP162, CEP90, MNR, or OFD1. Bottom panels show corresponding top views. Scale bar: 100 nm. **(B)** Quantification of top-view diameters for distal centriolar components: tubulin ($n = 15$, $171 \pm 14$ nm), NA14 ($n = 12$, $88 \pm 15$ nm), and C-terminal C2CD3 ($n = 18$, $76 \pm 14$ nm). Data from three independent experiments. **(C)** Axial positions of centriolar proteins relative to the distal tubulin signal: N-term C2CD3: $n = 42$, $-24 \pm 12$ nm; Middle C2CD3: $n = 37$ $-32 \pm 10$ nm; C-term C2CD3: $n = 33$, $-30 \pm 10$ nm; NA14: $n = 36$, $-30 \pm 13$ nm; SFI1: $n = 32$, $-30 \pm 13$ nm; CEP162: $n = 24$, $-4 \pm 19$ nm; CP110: $n = 47$, $28 \pm 12$ nm; CEP90: $n = 36$, $-56 \pm 17$ nm; MNR: $n = 56$, $-37 \pm 11$ nm; OFD1: $n = 46$, $-48 \pm 16$ nm. Data from three independent experiments. **(D)** Confocal images of expanded U2OS centrioles co-stained for tubulin (magenta) and dual-labeled (green) for CEP90–NA14, C-terminal C2CD3–CEP90, NA14–MNR, or C-terminal C2CD3–MNR. Right panels highlight the relatively proximal localization of CEP90 (white arrows). Scale bar: 100 nm. **(E)** iU-ExM image of centrioles stained for tubulin (magenta), middle C2CD3, and CEP90 (green), used for angle measurement (white lines). Scale bar: 100 nm. **(F)** Measured angles from (E). From 111 angles and 3 independent experiments. Average: $158°3 \pm 8.3$. **(G)** Schematic depicting CEP90 localization on the A-microtubule, consistent with observed angles. **(H)** Integrative model of distal centriole architecture: C2CD3 (green), NA14 (mustard), DISCO complex (black/yellow/orange), SFI1-C (orange), and Cep135 (dark blue), highlighting the proximity between C2CD3 and the DISCO complex. **(I–N)** Microtubule displacement assay with percentage of GFP/mCherry positive cells with proteins localized to cytosol (Cyt.) or microtubules (MT) for each condition. U2OS cells were transfected with (I) MNR-mCherry (Cyt.: 0; MT: 100), (J) C2CD3-GFP (Cyt.: 100; MT: 0), (K) C2CD3-GFP and MNR-mCherry (Cyt.: $64.23 \pm 11.02$), (L) C2CD3-GFP, MNR-mCherry and mCherry-OFD1 (Cyt.: $49.83 \pm 9.33$; MT: $50.17 \pm 9.33$), (M) C2CD3-GFP, MNR-mCherry and CEP90 (Cyt.: $8.254 \pm 7.211$; MT: $91.75 \pm 7.211$), N) C2CD3-GFP, MNR-mCherry, mCherry-OFD1 and CEP90 (Cyt.: $20.40 \pm 18.32$; MT: $79.60 \pm 18.32$). Note that C2CD3 co-transfection with MNR alone or in combination with OFD1 is not sufficient to completely drive its relocalization on MTs. C2CD3 is relocalized on MNR decorated microtubules most efficiently in combination with CEP90 and/or CEP90 and OFD1. Scale bar: 10 μm. From three independent experiments performed for each condition. **(O)** Model illustrating how C2CD3 extends between the microtubule triplets to connect with the DISCO complex. The detailed statistics of all the graphs shown in the figure are included in the S1 Data file.

From these findings, we hypothesized that C2CD3 might bridge the centriole lumen to the external DISCO complex, forming an in-to-out connection through the space between microtubule triplets/doublets (Fig 3H, model). To validate this model, we examined whether C2CD3 interacts with another component of the DISCO complex, particularly with MNR, a microtubule-associated protein positioned closest to the tubulin wall [13], and, therefore, a strong candidate for hook composition. We performed a displacement assay using MNR as bait (Fig 3I). First, we tested whether C2CD3 is able to associate with microtubules and found that, in contrast to previous reports, C2CD3 did not bind microtubules when overexpressed alone [16,31] (Fig 3J). However, when C2CD3 and MNR were co-expressed, 30% of cells show clear C2CD3 recruited to microtubules with little cytoplasmic background (Fig 3K), suggesting an interaction with MNR. Furthermore, as previously reported, we observed that MNR interacts with OFD1 and CEP90 independently (S5 Fig) [13]. Interestingly, overexpression of MNR with C2CD3, C2CD3/CEP90, and C2CD3/OFD1/CEP90 led to increased microtubule localization, suggesting a cooperativity in the binding (Figs 3L–3N and S5).

Together, our localization data, protein interaction assays, and cryo-ET observations support a model in which C2CD3 forms a radial structure linking the centriole lumen to the DISCO complex via an interaction with MNR (Fig 3O).

## C2CD3 acts as a molecular hub for the luminal ring and DISCO complexes

To better understand the functional roles of C2CD3 in organizing the distal centriole, we performed depletion experiments and analyzed their effects on the LDR and DISCO complexes.

First, we depleted C2CD3 and validated the efficiency of our siRNA by reproducing the previously reported formation of small centrioles without affecting the diameter (Fig 4A) [16]. Given that NA14 colocalizes with C2CD3, we also depleted it, expecting a similar phenotype. However, we observed only a mild reduction in centriole size and additionally a slight decrease in centriole diameter (Fig 4B). Next, to assess whether C2CD3 and NA14 are interdependent and whether the LDR complex was affected, we depleted C2CD3 or NA14 and analyzed the presence of SFI1, CEP135, and Centrin (Fig 4C–4J). We found that C2CD3 depletion resulted in the loss of all these components, including NA14, whereas NA14

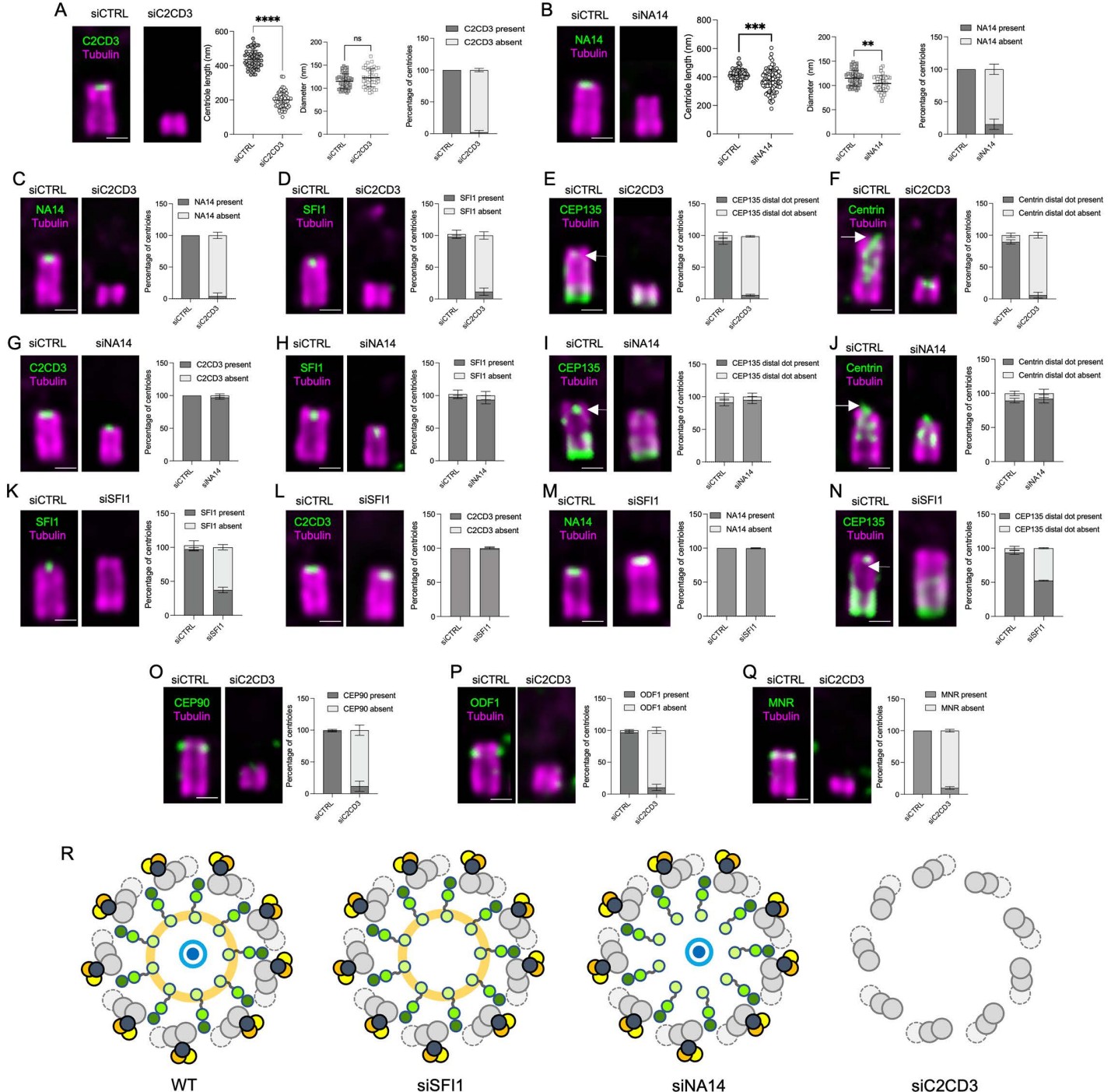

**Fig 4. C2CD3 is crucial for the localization of the LDR and DISCO complexes. (A)** Expanded centrioles from control (siCTRL) and C2CD3-depleted (siC2CD3) cells stained for C2CD3 (green) and tubulin (magenta). Centriole length is significantly reduced in siC2CD3 ($n = 61$, $200 \pm 48$ nm) compared to siCTRL ($n = 63$, $437 \pm 52$ nm); ****$P < 0.0001$, unpaired t *test*, 4 experiments. Centriole diameter was not significantly altered (siCTRL: $n = 44$, $115 \pm 24$ nm; siC2CD3: $n = 44$, $123 \pm 27$ nm; $P = 0.0564$). C2CD3 depletion efficiency: C2CD3-positive centrioles: siCTRL = 99.5% ± 1; siC2CD3 = 2.24% ± 2.66 ($n = 4$ experiments, 30 centrioles/experiment). **(B)** Expanded centrioles from siCTRL and NA14-depleted (siNA14) cells stained for NA14 (green) and tubulin (magenta). Centriole length is modestly reduced (siNA14: $n = 97$, $373 \pm 85$ nm; siCTRL: $n = 58$, $413 \pm 69$ nm; ***$P = 0.004$). Diameter is also decreased (siNA14: $n = 35$, $104 \pm 17$ nm; siCTRL: $n = 44$, $115 \pm 16$ nm; **$P = 0.0041$). NA14-positive centrioles: siCTRL = 100% ± 0; siNA14 = 15.38% ± 7.94 ($n = 4$ experiments). **(C–F)** Quantification of NA14, SFI1, CEP135 distal dot, and Centrin distal dot signals at centrioles in siCTRL and siC2CD3 cells. Percentage of NA14-positive and

NA14-negative centriole. For NA14-positive centrioles: siCTRL=100%±0, siC2CD3=4.03%±4.98. N=4 independent experiment (30 centrioles per experiment). Percentage of SFI1-positive and SFI1-negative centriole. For SFI1-positive centrioles: siCTRL=97.91%±2.55, siC2CD3=11.48%±5.77. N=3 independent experiment (30 centrioles per experiment). Percentage of CEP135 distal dot-positive and CEP135 distal dot-negative centriole. For CEP135 distal dot positive centrioles: siCTRL=93.65%±2.78, siC2CD3=5.91%±1.43. N=3 independent experiment (30 centrioles per experiment). Percentage of Centrin distal dot-positive and Centrin distal dot-negative centriole. For Centrin distal dot positive centrioles: siCTRL=89.48%±3.35, siC2CD3=7.83%±2.88. N=3 independent experiment (30 centrioles per experiment). (G–J) Quantification of C2CD3, SFI1, CEP135 distal dot, and Centrin distal dot signals in siCTRL and siNA14 cells. Percentage of C2CD3-positive and C2CD3-negative centriole. For C2CD3-positive centrioles: siCTRL=100%±0, siNA14=97%±2.64. N=3 independent experiment (30 centrioles per experiment). Percentage of SFI1-positive and SFI1-negative centriole. For SFI1-positive centrioles: siCTRL=97.91%±2.50, siNA14=93.6%±6.25. N=3 independent experiment (30 centrioles per experiment). Percentage of CEP135 distal dot-positive and CEP135 distal dot-negative centriole. For CEP135 distal dot positive centrioles: siCTRL=91.31%±5.20, siNA14=95%±5.57. N=3 independent experiment (30 centrioles per experiment). Percentage of Centrin distal dot-positive and Centrin distal dot-negative centriole. For Centrin distal dot positive centrioles: siCTRL=89.48%±3.35, siNA14=92.15%±6.08. N=3 independent experiment (30 centrioles per experiment). (K–N) Quantification of SFI1, C2CD3, NA14, and CEP135 signals in siCTRL and siSFI1 cells. Percentage of SFI1-positive and SFI1-negative centriole. For SFI1-positive centrioles: siCTRL=97.22%±2.54, siSFI1=37.33%±4,07. N=3 independent experiment (30 centrioles per experiment). Percentage of C2CD3-positive and C2CD3-negative centriole. For C2CD3-positive centrioles: siCTRL=100%±0, siSFI1=98.83%±2.02. N=3 independent experiment (30 centrioles per experiment). Percentage of NA14-positive and NA14-negative centriole. For NA14-positive centrioles: siCTRL=100%±0, siSFI1=99.67%±0.57. N=3 independent experiment (30 centrioles per experiment). Percentage of CEP135 distal dot-positive and CEP135 distal dot-negative centriole. For CEP135 distal dot positive centrioles: siCTRL=93.65%±2.78, siSFI1=52.42%±0.76. N=3 independent experiment (30 centrioles per experiment). (O–Q) Quantification of CEP90, OFD1, and MNR signals at centrioles in siCTRL and siC2CD3 cells. Percentage of CEP90-positive and CEP90-negative centriole. For CEP90-positive centrioles: siCTRL=98.47%±1.36, siC2CD3=11.8%±8.07. N=3 independent experiment (30 centrioles per experiment). Percentage of OFD1-positive and OFD1-negative centriole. For OFD1-positive centrioles: siCTRL=96.73%±1.20, siC2CD3=10.32%±4.94. N=3 independent experiment (30 centrioles per experiment). Percentage of MNR-positive and MNR-negative centriole. For MNR-positive centrioles: siCTRL=100%±0, siC2CD3=10%±2.19. N=3 independent experiment (30 centrioles per experiment). (R) Schematic model summarizing the dependency of distal centriolar components on C2CD3. All centrioles are stained for tubulin (magenta) and the indicated protein (green). Scale bars: 200 nm. The detailed statistics of all the graphs shown in the figure are included in the S1 Data file.

depletion had no impact on the LDR, suggesting that C2CD3 is essential for its assembly and that NA14 may function downstream (Fig 4C and 4G–4J).

Next, we tested the interdependence between SFI1 and C2CD3. By depleting SFI1, we obtained a specific loss of the inner LDR proteins CEP135, Centrin-2, and SFI1 itself, while C2CD3 and NA14 (which localize to the ring structure), remained intact (Fig 4K–4N). This finding suggests that SFI1 plays a role in recruiting or stabilizing CEP135 within the luminal ring, but it is not required for the overall maintenance of the structure. The persistence of C2CD3 in SFI1-depleted cells further supports the idea that C2CD3 acts upstream in organizing the luminal ring and serves as a central anchoring point.

Finally, consistent with our C2CD3 "in-to-out" hypothesis, we investigated whether the previously reported loss of centriolar appendages in C2CD3 mutants [14,16] could be attributed to the disruption of the DISCO complex, which is known to act as an early anchoring platform for the recruitment of distal appendages [13,10]. As predicted, we observed an absence of the DISCO complex components at the centrioles of cells depleted of C2CD3, comprising CEP90, MNR, and OFD1 (Fig 4O–4Q), supporting the notion that C2CD3 is essential for stabilizing this complex.

In summary, our depletion experiments reveal a hierarchical organization of both the LDR and DISCO complexes (Fig 4R). These results strongly indicate that C2CD3 serves as a molecular hub of distal centriole architecture, linking the LDR to the more external DISCO complex and appendages that it recruits.

## C2CD3 controls the spatial organization of centriole architecture

C2CD3 depletion has been previously shown to result in shorter centrioles, and to impair distal appendage proteins and proper ciliogenesis [14,16]. Our findings now elucidate the molecular basis for the distal appendage defects by demonstrating that C2CD3 serves as a critical link between the centriole lumen and the DISCO complex, which is essential for distal appendage recruitment. However, these observations do not fully account for the shorter centriole phenotype, suggesting that C2CD3 may play an additional a role in organizing other structural features of the centriole. Therefore, we used U-ExM to analyze the recruitment of the cartwheel, the A-C linker, the inner scaffold, and the torus upon C2CD3 depletion by siRNA (Fig 5).

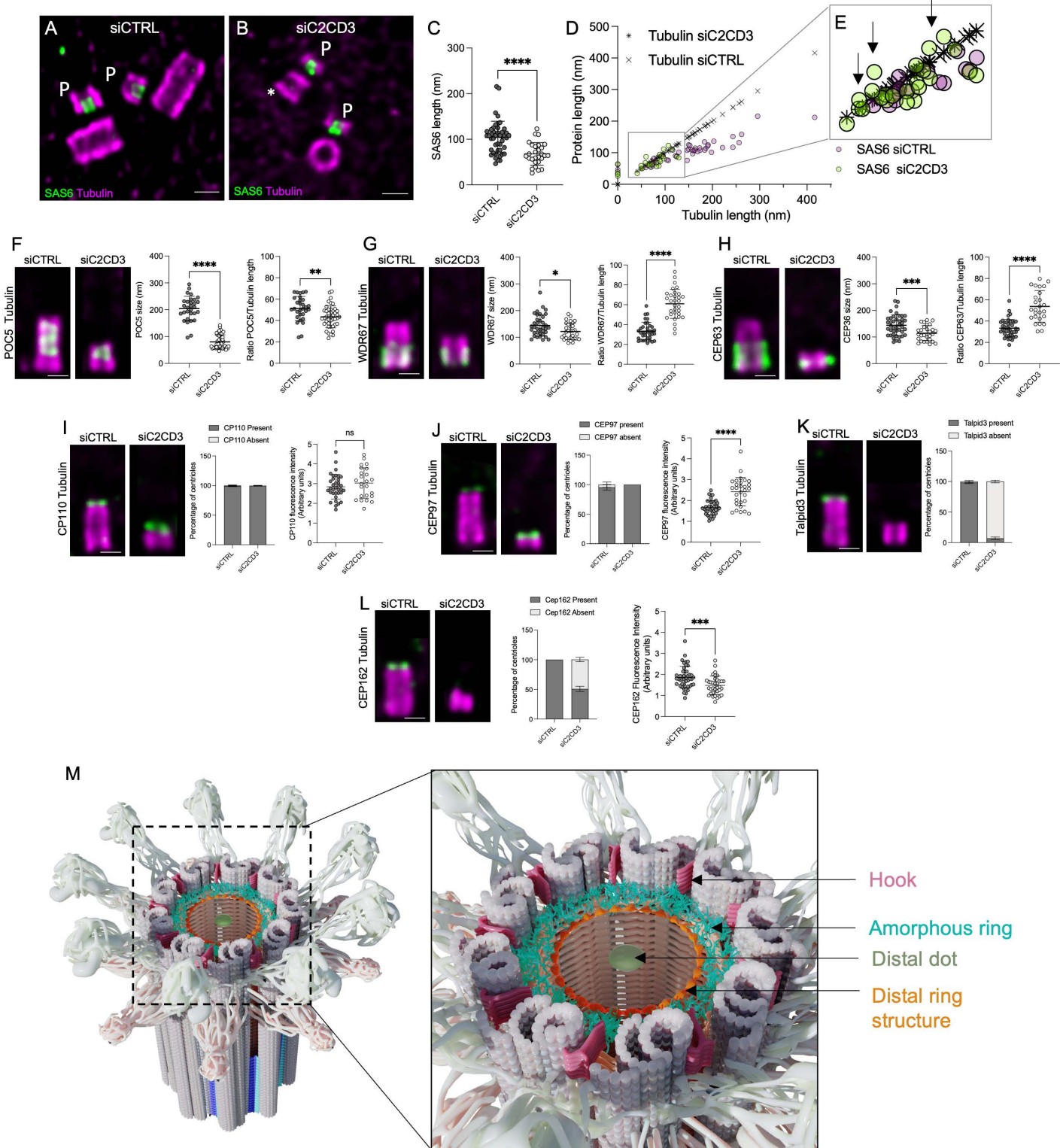

**Fig 5. C2CD3 regulates spatial organization of centriole architecture. (A, B)** Expanded U2OS centrioles treated with siCTRL or siC2CD3, stained for α/β-tubulin (magenta) and SAS6 (green); P = procentrioles, * = C2CD3-depleted. Scale bars: 200 nm. **(C)** SAS6 length is significantly reduced in siC2CD3 ($n = 28$, $68 \pm 25$ nm) vs. siCTRL ($n = 45$, $104 \pm 36$ nm), ****$P < 0.0001$. **(D, E)** Correlation of SAS6 and tubulin lengths. In siC2CD3, the cartwheel

length often exceeds tubulin length (back arrows). **(F)** Expanded centrioles stained for α/β-tubulin and the inner scaffold protein POC5. POC5 length: siC2CD3 ($n=43$, $80\pm26$ nm) vs. siCTRL ($n=30$, $204\pm46$ nm), ****$P<0.0001$. POC5-Tubulin length ratio of expanded centrioles in siCTRL (dark gray, $n=30$ centrioles; $51\pm11$ nm) and siC2CD3 (light gray, $n=42$, $43\pm10$ nm). Unpaired $t$ test ** $P$ value $=0.0030$. **(G)** WDR67 localization in expanded centrioles. WDR67 length: siC2CD3 ($n=31$, $122\pm31$ nm) vs. siCTRL ($n=35$, $145\pm40$ nm), *$P=0.0128$. WDR67-Tubulin length ratio of expanded centrioles in siCTRL (dark gray, $n=35$, $33\pm9$ nm) and siC2CD3 (light gray, $n=31$, $61\pm15$ nm). Unpaired $t$ test **** $P$ value $<0.0001$. **(H)** CEP63 localization in expanded centrioles. CEP63 length: reduced in siC2CD3 ($n=26$, $112\pm29$ nm) vs. siCTRL ($n=42$, $144\pm39$ nm), ***$P=0.0008$. CEP63-Tubulin length ratio of expanded centrioles in siCTRL (dark gray, $n=42$, $32\pm8$ nm) and siC2CD3 (light gray, $n=26$, $54\pm15$ nm). Unpaired $t$ test **** $P$ value $<0.0001$. **(I)** CP110 localization in expanded centrioles and quantification. Percentage of CP110-positive and CP110-negative centrioles. For CP110-positive centrioles: siCTRL $=99.33\%\pm1.15$, siC2CD3 $=99.67\%\pm0.58$. CP110 intensity doesn't increase in siC2CD3 ($n=24$, $3.02\pm0.77$ AU) vs. siCTRL ($n=35$, $2.84\pm0.62$ AU), **$P=0.0025$. **(J)** CEP97 localization in expanded centrioles and quantification. For CEP97-positive centrioles: siCTRL $=95.47\%\pm4.55$, siC2CD3 $=100\%\pm0$. CEP97 fluorescence increased in siC2CD3 ($n=29$, $2.42\pm0.68$ AU) vs. siCTRL ($n=38$, $1.65\pm0.35$ AU), ****$P<0.0001$. **(K)** TAL-PID3 localization in expanded centrioles and quantification. For TALPID3-positive centrioles: siCTRL $=98.3\%\pm1.57$, siC2CD3 $=6.95\%\pm2.09$. TALPID3 is lost in siC2CD3 vs. siCTRL. **(L)** CEP162 localization in expanded centrioles and quantification. Half of the centrioles lacked detectable CEP162. For CEP162-positive centrioles: siCTRL $=100\%\pm0$, siC2CD3 $=50.83\%\pm4.31$. Overall CEP162 fluorescence decreased in siC2CD3 ($n=31$, $1.47\pm0.39$ AU.) vs. siCTRL ($n=35$, $1.68\pm0.51$ AU), unpaired $t$ test *** $P$ value $=0.0008$. **(M)** Model of the architecture of the distal region of the human centriole. All data are from three independent experiments unless stated otherwise. All centrioles are stained for tubulin (magenta) and the indicated protein (green). Scale bars: 200 nm. The detailed statistics of all the graphs shown in the figure are included in the S1 Data file.

First, we assessed centriole duplication and found no significant defects in procentriole formation, despite a slight reduction in S-phase cells and overall cell count after siRNA treatment (Figs 5A and S6). We examined cartwheel assembly using SAS-6 as a marker (Fig 5A and 5B). While SAS-6 was recruited to centrioles, the procentriole and cartwheel appeared shorter (Fig 5C). Specifically, by plotting the length of sas-6 as a function of the tubulin length of mature centrioles and procentrioles, we observed an abnormal proportion of procentriole with cartwheels exceeding their associated microtubules in length (Figs 5D, 5E, and S6E). These results suggest that C2CD3 depletion may impact early centriole assembly dynamics or microtubule growth, resulting in shorter cartwheels and centrioles without blocking duplication.

We then evaluated the recruitment of other structural components, including the inner scaffold (POC5), the A-C linker (WDR67), and the torus (CEP63). We found that while these proteins were still recruited to centrioles, their signal lengths were significantly reduced, consistent with the smaller centriole size (Fig 5F–5H). To investigate whether these reductions were proportional to the centriolar length, we compared their signal lengths to that of tubulin. Interestingly, while the inner scaffold (POC5) was slightly shorter, the A–C linker and torus appeared proportionally longer relative to the tubulin signal (Fig 5F–5H). This indicates that, although these structures are recruited, their spatial organization is disrupted in C2CD3-depleted centrioles.

Finally, we explored whether the reduction in centriole size could be linked to impaired microtubule triplet elongation, which is regulated by distal cap proteins such as CP110 and CEP97 [28,32]. In parallel, we also tested other distal cap proteins TALPID3 and CEP162. Upon C2CD3 depletion, TALPID3 and CEP162 were completely lost or reduced, while CP110 and CEP97 retained their localization. Interestingly, fluorescence intensity measurements revealed that CP110 levels remained unchanged, whereas CEP97 was increased (Fig 5I and 5J). Given that CEP97 is a known regulator of centriolar microtubule growth and inhibits microtubule dynamics in both human and Drosophila systems [33], its upregulation may contribute to the shorter centriole phenotype observed.

In summary, our results demonstrate that C2CD3 not only orchestrates the recruitment and assembly of the LDR and DISCO complexes but also regulates the localization and abundance of distal cap proteins. These findings establish C2CD3 as a master organizer of the molecular architecture of the centriole's distal end, coordinating both assembly and structural integrity.

## Discussion

Our findings establish C2CD3 as a central architectural organizer of the distal centriole, bridging the LDR complex and peripheral appendage assembly sites (Fig 5M). We show that the LDR, and in particular C2CD3 and NA14, are part of a

poorly understood structural element of the centriole, a luminal ring that resembles a "pearl necklace." C2CD3 arranges in a 9-fold radially symmetric pattern reminiscent of the proximal cartwheel structure [34], particularly echoing the oligomeric form of SAS-6, which assembles into a~25 nm central hub with nine spokes extending toward the microtubule triplets [4,35–37]. This architectural similarity suggests an evolutionarily conserved principle of radial organization at both ends of the centriole, with C2CD3 potentially serving a cartwheel-like scaffolding function at the distal end. However, the distal ring structure is markedly different: it forms a ring approximately 100 nm in diameter, composed of 27 distinct nodes (or "pearls"), with no discernible spokes. Instead, an amorphous region is observed, along with hook-like structures between each microtubule triplet. The presence of 27 nodes suggests a triplet of nodes per each of the centriole's nine asymmetric units. Given the multiple C2 domains present in C2CD3, these nodes could correspond to individual domains. However, considering that a single C2 domain has a molecular weight of ~18 kDa, which would not produce a density of the size seen by cryo-ET, it is more plausible that each node represents a dimer or trimer of C2 domains. Interestingly, only the C-terminal region of C2CD3 (with only one C2 domain) localizes to the distal ring structure, raising the possibility that these nodes may correspond to a different protein. NA14 is a strong candidate in this context, as it precisely co-localizes with the ring structure (Fig 3H). Although NA14 is too small (~14 kDa) to form a node as a monomer, its known oligomerization properties suggest potential functional complexity [38]. However, our iU-ExM data also showed a 9-fold symmetrical arrangement of NA14 (rather than 27-fold), leaving open the possibility that other proteins may contribute to the formation of the nodes.

We propose that the amorphous region is at least partially composed of C2CD3, which, even when organized with 9-fold symmetry, adopts a more flexible architecture extending from the luminal ring structure to the hook structure. Notably, the hook is positioned close to the N-terminus of C2CD3, and previous studies suggested that C2CD3 interacts with microtubules [16,39]. Although we could not replicate this interaction under our experimental conditions, possibly due to regulatory influences such as expression levels or cell cycle stage, it remains plausible that the N-terminal domain of C2CD3 contributes to the hook structure or interacts transiently with microtubules. This idea is further supported by the interaction of C2CD3 with the DISCO complex, located on the outer surface of triplet microtubules.

Interestingly, we observed a longitudinal offset between the hook structure and the DISCO complex. While the hook is aligned with and extends distally from the ring structure, the DISCO complex is positioned more proximal. This spatial arrangement suggests that the hook is not composed of the DISCO complex, but rather of other, as yet unidentified, proteins. Moreover, the distal extension of the hook structure does not perfectly match the localization of the C2CD3 N-terminus, which is strictly confined to the plane of the ring. Future high-resolution structural studies of the hook, as well as identification of novel C2CD3 interactors, will be crucial to refine this emerging model of distal centriole architecture.

We demonstrated the presence of the distal luminal ring structure in specialized primary cilia of photoreceptor cells, as well as in multiciliated cells of the mouse airway. This structure was previously reported in oviduct epithelium [20], pig kidney [21], and HeLa cells [22,23], suggesting its conservation in centrioles across metazoans. However, its evolutionary conservation across all centriole-bearing organisms remains unclear. To date, there are no reports of such a ring structure in other species. Interestingly, it has been proposed that the C2CD3 ring is reminiscent of the "Acorn" structure observed in the green alga *Chlamydomonas reinhardtii* [24]. Although the Acorn is asymmetrical and lacks 9-fold symmetry [40], it shares some molecular similarities. For instance, while no clear ortholog of C2CD3 has been identified in *C. reinhardtii*, centrin is present in the LDR and localizes very close to the Acorn [41]. This raises the possibility that the Acorn and luminal ring structures share an ancestral origin but diverged functionally and structurally across species.

Another line of evidence supporting conservation comes from the presence of a functional C2CD3/NA14 complex in the worm *Caenorhabditis elegans*, where the orthologs SAS-1 and SSNA-1 genetically interact [42]. SSNA-1 is essential for centriole integrity after assembly and has been implicated in both centriole stability and ciliary function in *C. elegans* [43] and human cells [44]. Additionally, NA14 is conserved in *C. reinhardtii*, localizing to the basal body/flagella complex [45], and is also found at the conoid in *Toxoplasma gondii* [46], a tubulin-based structure thought to have evolved from the

centriole/flagella complex [47]. Thus, systematic exploration of C2CD3 and SSNA-1 conservation across the tree of life could reveal whether the LDR structure originated in the Last Eukaryotic Common Ancestor or represents a more recent evolutionary innovation in certain opisthokonts.

Both C2CD3 and NA14 are implicated in severe ciliopathies. C2CD3 is essential for craniofacial development and skeletal patterning, and its dysfunction leads to skeletal dysplasia and oral-facial-digital (OFD) syndromes, ciliopathies marked by defects in the face, oral cavity, and limbs [14,16,48–50]. NA14 (also known as SSNA-1 due to its role in primary Sjögren's syndrome, a chronic autoimmune disease [51]) has not been directly classified as a ciliopathy-associated protein. However, this may reflect its multifunctionality. Regardless, the central role of C2CD3 in organizing the distal centriole architecture likely underlies its widespread involvement in ciliopathies. Understanding its molecular organization may reveal that these disorders are not only due to the loss of distal appendages but may also stem from more subtle architectural defects that impair diverse cellular processes.

Finally, our study highlights the power of multiscale imaging and the complementary strengths of in situ cryo-ET and U-ExM. This combination has already enabled the discovery of key features of centriole molecular architecture [7,52–54] and even cytoskeleton components in archaeal cells [55]. Cryo-ET reveals intricate native structural details in a more restricted field of view than U-ExM, but subtomogram averaging is limited by low particle numbers and intrinsic heterogeneity, which hinders atomic resolution and de novo protein identification. In contrast, U-ExM allows precise protein localization within architectural elements but lacks direct structural context. A limitation of U-ExM is dimensional variability, likely due to gel inhomogeneity. However, the centriole is an excellent nanoruler: by labeling tubulin and measuring centriole diameter, U-ExM data can be rescaled to match cryo-ET measurements, which are more consistent and unaffected by expansion artifacts. This makes centrioles an excellent model for such integrative multiscale imaging, an approach that could be extended to other subcellular structures, ultimately revealing the architecture of organelles across cell types and organisms [56].

Together, these insights significantly expand our understanding of centriole architecture and maturation, and position C2CD3 as a multifunctional scaffold with relevance to both centriole biology and ciliopathy mechanisms.

## Materials and methods

### Ethics statement

All animal procedures for the cryo-focused ion beam (cryo-FIB) preparation of mouse trachea were approved by the Stanford University Administrative Panel on Laboratory Animal Care (SUAPLAC; protocol #11659) and conducted in accordance with institutional and national guidelines for animal welfare. Experiments involving the cryo-FIB of mouse retina and U-ExM of mouse trachea were performed in compliance with the ethical standards set forth in the Swiss Federal Act on Animal Protection and the Swiss Animal Protection Ordinance. These protocols were reviewed and approved by the University of Geneva and the Canton of Geneva ethics committees (authorization GE279/ National number: 34121).

### Human cell lines and cell culture

U2OS (human osteosarcoma cells; ATCC-HTB-96) cells were cultured in DMEM medium supplemented with GlutaMAX (Life Technology), 10% tetracycline-negative fetal calf serum (Life Technology), penicillin, and streptomycin (100 μg/ml) at 37 °C and 5% $CO_2$. Cells were regularly tested for mycoplasma contaminations.

### C2CD3, NA14, and SFI1 depletion using siRNAs

The siRNAs used to deplete C2CD3, NA14, and SFI1 were purchased from ThermoFisher. The sequences are as follows: siC2CD3#1 ref sequence NM_015531 and ref siRNA s229844 (sense sequence CGAUGACACUAAGUGUGGATT). siNA14#1 ref sequence NM_003731 and ref siRNAi s194945 (sense sequence ACGAGAACCUGGCACGCAATT) and

siNA14#2 ref sequence NM_003731 and ref siRNAi s228421 (sense sequence CUCGCUUCAUGCUCACACATT). The siRNAs and conditions used to deplete SFI1 were previously reported in [30]. Silencer select negative control siRNA1 were purchased from ThermoFisher (4390843, ThermoFisher).

U2OS cells were plated (100'000 cells) on 12- or 15-mm coverslips in a 6-well plate and 40 nM siRNA-C2CD3 or a mix of siRNA-NA14-1 and siRNA-NA14-2 (20 nm each) were transfected using RNAi MAX reagents (Invitrogen) according to the manufacturer protocol and medium was changed 6 hours after transfection. Cells were analyzed 72 hours after transfection.

Note that C2CD3 protein levels could not be assessed by western blot due to the lack of detectable signal under control conditions.

## Antibodies

The primary antibodies used in this study are described in the following table.

| Protein | Reference | Company | Dilution for IF | Dilution for U-ExM | Dilution for iU-ExM |
|---|---|---|---|---|---|
| NA14 (SSNA1) | 11797-1-AP | Proteintech | NA | 1/250 | 1/250 |
| C2CD3-Cterm | HPA038552 | Atlas Antibody | NA | 1/250 | 1/250 |
| α−Tubulin | AA345, scFv-F2C | ABCD Antibodies | 1/1000 | 1/125 | 1/250 |
| β−Tubulin | AA344, scFvS11B | ABCD Antibodies | 1/1000 | 1/125 | 1/250 |
| C2CD3-Nterm | af7348 | R&D systems | NA | 1/125 | 1/250 |
| C2CD3-Middle | HPA040433 | Atlas Antibody | NA | 1/250 | 1/250 |
| SFI1 | 13550-1-AP | Proteintech | 1/500 | 1/250 | |
| CP110 | 12780-1-AP | Proteintech | 1/500 | 1/250 | |
| CEP162 | HPA030173 | Atlas Antibodies | NA | 1/250 | |
| CEP90 (PIBF1) | 144-1-AP | Proteintech | NA | 1/250 | 1/250 |
| MNR (KIAA0753) | NBP1-90929 | Novusbio | 1/1000 | 1/250 | 1/250 |
| OFD1 | HPA031103 | Atlas Antibody | 1/1000 | 1/250 | |
| GFP | MAB3580 | Merk | 1/500 | 1/250 | |
| mCherry | ab167453 | Abcam | 1/250 | 1/250 | |
| CEP135 | 24428-1-AP | Proteintech | NA | 1/250 | |
| Centrin (clone 20H5) | 04-1624 | Millipore | 1/500 | 1/250 | |
| Hs-SAS6 (SAS-6) | sc-81431 | Santa Cruz Biotechnology | NA | 1/250 | |
| Talpid3 | 24421-1-AP | Proteintech | NA | 1/250 | |
| CEP63 | 16268-1-AP | Proteintech | NA | 1/250 | |
| WDR67 | HPA023710 | Atlas antibody | NA | 1/250 | |

Secondary fluorescent antibodies were purchased from Invitrogen (A11008, A11004, A11029, A11036, A11075), Jackson Immuno (06-175-148), and used at 1:800 dilutions for standard immunofluorescence experiments and 1:400 for U-ExM.

## Cloning

The following constructs used in this work were previously described: MNR-GFP, OFD1-mCherry, and CEP90 [34] (gifts from Anne-Marie Tassin, France); C2CD3-GFP [4] (gift from Pierre Gönczy, Switzerland); and SFI1 [33]. Plasmids encoding MNR-mCherry and mCherry-OFD1 were kindly provided by Olivier Rosnet (France).

## Displacement assay and immunofluorescence

U2OS cells were grown in Dulbecco's modified Eagle's medium (DMEM) supplemented with 10% fetal calf serum (Life Technologies) and penicillin and streptomycin (100 μg/ml). U2OS cells grown on coverslips were transfected at 80% confluency in a six-well plate using jetPRIME reagent (Polyplus-transfection) following the manufacturer's

instructions, with 2.5 µg of total DNA of the following combinations: C2CD3-GFP alone, MNR-mCherry alone, CEP90 alone, mCherry-OFD1 alone, C2CD3-GFP with either MNR-mCherry, CEP90, or mCherry-OFD1, CEP90 with MNR-GFP, mCherry-OFD1 with MNR-GFP, C2CD3-GFP with MNR-mCherry and CEP90, C2CD3-GFP with MNR-mCherry and mCherry-OFD1, C2CD3-GFP with MNR-mCherry, mCherry-OFD1 and CEP90. The medium was changed after 4–6 hours after transfection and doxycycline (100 ng/ml) was added to the combinations containing C2CD3-GFP. The expression of fluorescent fusion proteins was allowed for 24 hours. Cells were then fixed in −20 °C cold MeOH for 3 min and washed with PBS three times. Cells were then stained in PBS-BSA 2% with a mix of α- and β-tubulins (α-Guinea Pig, AA345 and AA344 from ABCD antibodies, both diluted at 1:250) and CEP90 (Rabbit, Proteintech 14413-1-AP) in transfections where CEP90 staining was shown. Primary antibody staining was followed by anti-Rabbit IgG, Alexa Fluor 568 (Invitrogen, A11036) and/or anti-Guinea Pig IgG, Cy5 (Jackson ImmunoResearch, 706-175-148), washed three times in PBS Tween 0.1%. Coverslips were mounted using glycerol-mounting medium with 4′,6-diamidino-2-phenylindole and DABCO 1, 4-diazabicyclo (2.2.2) octane (Abcam, ab188804). Images were acquired and processed using Leica Thunder microscope.

## Ultrastructure expansion microscopy (U-ExM)

U2OS cells were processed for U-ExM protocol as previously described [7,57,58]. The following reagents were used in U-ExM experiments: formaldehyde (FA, 36.5%–38%, F8775, SIGMA), acrylamide (AA, 40%, A4058, SIGMA), N, N′-methylenbisacrylamide (BIS, 2%, M1533, SIGMA), sodium acrylate (SA, 97%–99%, 408220, SIGMA), ammonium persulfate (APS, 17874, ThermoFisher), tetramethylethylendiamine (TEMED, 17919, ThermoFisher), nuclease-free water (AM9937, Ambion-ThermoFisher) and poly-D-Lysine (A3890401, Gibco).

Briefly, U2OS cells were grown on 12 mm coverslips and processed the day after. Coverslips were incubated in 2% AA + 1.4% FA diluted in PBS for 3 hours at 37 °C prior to gelation in monomer solution (19% SA, 0.1% BIS, 10% acrylamide) supplemented with TEMED and APS (final concentration of 0.5%) between 30 min and 1 hour at 37 °C. Denaturation was performed for 1h30 at 95 °C and gels were stained as described above. All dimensional measurements were corrected according to the calculated expansion factor.

## Image acquisition and analysis

Expanded gels were mounted on 24 mm coverslips pre-coated with poly-D-lysine (0.1 mg/mL) and imaged using an inverted Leica TCS SP8 or Stellaris 8 confocal microscope, or an inverted Leica Thunder widefield system. For widefield imaging, a 63 × 1.4 NA oil immersion objective was used in "Small Volume Computational Clearing" mode with water as the mounting medium to generate denoised images. 3D image stacks were acquired at 0.21 µm z-intervals with an xy pixel size of 100 nm. Confocal imaging on the SP8 and Stellaris microscopes was performed with a 63 × 1.4 NA oil objective using Lightning mode at maximum resolution, with "Adaptive" selected as the deconvolution strategy and water as the mounting medium. 3D stacks were acquired at 0.12 µm z-intervals with a lateral pixel size of 35 nm.

Centriole length and diameter quantifications were performed as previously described in [7] using two custom ImageJ plugins detailed in [59]. For protein intensity measurements in U-ExM images, a 20 × 20-pixel square region of interest (ROI) was drawn on a Z projection of 3 consecutive slides, and raw integrated fluorescence intensity was measured. Background intensity was determined from a nearby region and subtracted from the ROI signal before plotting.

siRNA knockdown efficiency was evaluated manually in G1-phase cells (identified by the presence of two centrioles). The presence or absence of the target protein at the centrioles was used to classify cells into two categories.

## Mouse trachea tissue preparation for U-ExM

Mouse tracheas were collected and incubated 30 min in 2% PFA in PEM buffer at room temperature (100 mM PIPES, 5 mM EDTA, 5mM $MgCl_2$, pH 6.9 (adjusted with KOH) in $H_2O$). Then, tracheas were included in OCT mounting medium

into cryomolds (#361603E, VWR) by carefully orienting them for future sectioning. Cryomolds were then frozen by putting them into a Becher filled with isopentane, itself cooled down thanks to liquid nitrogen. Frozen cryomolds are then stored at −80 °C. From these blocks, 10 μm thick cryosections were cut with a cryostat and put on slide where expansion microscopy was performed as described here [60].

### Iterative-ultrastructure expansion microscopy (iU-ExM)

U2OS cells were processed using the iterative expansion microscopy protocol as previously described [25]. Briefly, cells were incubated in a solution of 1.4% FA and 2% acrylamide (AAm) in 1× PBS for 3 hours at 37 °C. Coverslips were then sealed in gelation chambers placed on ice in a humidified environment and filled with a monomer solution containing 10% AAm, 19% SA, 0.1% DHEBA, and 0.25% APS/TEMED. After 15 min on ice, the chambers were transferred to 37 °C for 45 min to complete gelation. The coverslip with the gel was immersed in iU-ExM denaturation buffer (200 mM SDS, 200 mM NaCl, 50 mM Tris base, pH 6.8) in a 6-well plate until the gel detached. It was then transferred to a 1.5 mL tube containing fresh denaturation buffer and incubated at 85 °C for 1.5 hours. Following denaturation, the gel underwent initial expansion in three 30-min washes with ddH$_2$O. After the first expansion, the gel was immunostained and re-embedded for the second expansion. A ~1.5 × 1.5 cm piece was placed in a 6-well plate with 8 mL of neutral gel monomer solution (10% AAm, 0.05% DHEBA, 0.05% APS/TEMED) on ice with gentle shaking for 30 min. The gel was then mounted on a microscope slide, excess monomer solution was removed using Kimwipes, and a 22 × 22 mm coverslip was placed on top. Remaining space was filled with monomer solution, and the slide was incubated in a humid chamber at 37 °C for 1 hour.

The neutral gel-embedded sample was then incubated in anchoring solution (1.4% FA/ 2% AAm) for 3–5 hours at 37 °C with shaking, followed by a 30-min wash in 1× PBS. Subsequently, the gel was incubated in second expansion monomer solution (10% AAm, 19% SA, 0.1% BIS, 0.05% APS/TEMED) for 30 min on ice with shaking. After removing excess solution with Kimwipes, the gel was covered with a coverslip, and the space was filled with remaining monomer solution. Polymerization was completed in a humid chamber at 37 °C for 1 hour. Following polymerization, the gel was incubated in 200 mM NaOH for 1 hour at room temperature with shaking, then washed in 1× PBS (~20 min per wash) until pH reached 7. Final expansion was achieved by incubating the gel in ddH$_2$O, changing the water until the gel expansion plateaued. Note that for samples with weak signal, we applied a triple-round labeling strategy referred to as Fluoboost (Louvel and colleagues, in preparation). Gels were incubated with primary antibodies at 1:250 dilution (See Antibodies) for 2h30 hours, washed 3 × 10 min in PBS + 0.1% Tween-20 (PBS-T), then incubated with an unlabeled secondary antibody (1:250) for 1.5 hours at 37 °C with shaking. After another 3 × 10 min PBS-T wash, gels were incubated with a tertiary, fluorophore-conjugated antibody (1:400) for 2 hours at 37 °C, followed by final PBS-T washes and re-expansion in ddH$_2$O. The following unlabeled secondary antibodies were used in this study: chicken anti-mouse (ThermoFisher, A15977) and rabbit anti-sheep (ThermoFisher, AB_228453). All dimensional measurements were corrected according to the calculated expansion factor.

### Serial lift-out, cryo-FIB-milling, and cryo-tomography of mouse retina

Retinas from C57BL/6J WT mice (>4 weeks old) were dissected as previously published [24], and stained with Hoechst (20 μM) in Ringer's buffer for targeting. After incubation in 20% dextran cryoprotectant, samples were high-pressure frozen in 200 μm planchettes using a Leica EM ICE system. Cryo-trimming (~30–50 μm) was performed at −170 °C using a Leica UC6-FCS cryomicrotome and a Diatome Trim 20 diamond knife. Trimmed planchettes were imaged using a Leica Cryo-Thunder epifluorescent microscope. Regions of interest were identified and mapped relative to tissue. Targeted samples were then transferred to a Thermo Scientific Aquilos 2 cryo-FIB-SEM for lamella preparation following the SOLIST workflow [61], with sequential thinning and final polishing to ~150–200 nm thickness [62].

Five cryo-ET was performed on a Titan Krios G4 microscope with SelectrisX filter and Falcon 4i detector. Tilt series (−50° to +70°, 2° steps) were collected using a dose-symmetric scheme (total ~180 e⁻/Å², pixel size 2.42 Å, in EER mode). WarpTools was used for processing; EER frames were grouped to ~0.25–0.3 e⁻/Å² per frame. Final tomograms were denoised using cryoCARE [63] for enhanced contrast and structural visibility. Of the five tomograms collected, two displayed a clearly resolved ring structure.

## Cryo-FIB-milling and cryo-tomography of primary mouse tracheal epithelial cells

Primary MTECs were isolated from wild-type CD-1 or GFP-centrin2 transgenic mice (University of Minnesota). Following euthanasia by CO₂ asphyxiation, tracheae from approximately 10 mice were dissected, opened longitudinally, and stored in ice-cold PBS. The tissue was then incubated in Pronase E (in DMEM:F12 with antibiotics/antimycotics) at 4 °C overnight (~18 hours) to dissociate epithelial and basal cells from the extracellular matrix. Enzymatic activity was quenched by adding FBS to a final concentration of 10%. The tracheae were then vigorously agitated to release the cells, which were collected using a Pasteur pipette. Empty tissue remnants were discarded, and the pooled cell suspensions were centrifuged at 600g for 10 min at 4 °C. The resulting pellet was resuspended in DNase and incubated on ice for 5 min before another spin at 300g for 10 min. Cells were then resuspended in 10% FBS in DMEM:F12 AB/AM and incubated for 4 h at 37 °C and 5% CO₂ in a 10 cm Primaria plate to promote adhesion.

After centrifugation at 400g for 10 min at room temperature, the cell pellet was resuspended in CM + RA medium. Collagen-coated Corning Transwell inserts (6- or 24-well format) were rinsed with CM + RA and seeded with 1 mL (6-well) or 250 μL (24-well) of the cell suspension. CM + RA (1.5 mL or 0.5 mL, respectively) was added to the basolateral chamber. Cultures were incubated undisturbed at 37 °C and 5% CO₂ for 5–6 days, with media refreshed every other day. ROCK inhibitor was included in CM + RA until day 4. By day 5–6, confluency and epithelial morphology were assessed by light microscopy. When a compact, columnar cell layer had formed, apical medium was removed to initiate air–liquid interface (ALI) conditions, and the basolateral medium was replaced with Nu + RA or SF + RA. Media changes continued every other day, with apical surfaces rinsed using ~1.5 mL PBS to remove mucus.

From day 7 to day 30 of ALI culture, cells were harvested for analysis. Wells were rinsed twice with BM and treated with 1.25% trypsin (0.5 mL basolateral, 0.25 mL apical) overnight at 37 °C and 5% CO₂. The dissociation was quenched with 0.25 mL FBS, yielding a 1 mL single-cell suspension. Cells were counted using a hemocytometer and monitored for ciliary motility, then centrifuged at 600g for 10 min. The pellet was resuspended in BM + 10% FBS at the desired concentration for EM grid application.

A 4 μL aliquot of cell suspension was mixed with 1 μL of 10- or 15-nm BSA-coated gold fiducials and applied to carbon-coated 200-mesh copper EM grids (Quantifoil). The grids were plunge-frozen into liquid ethane using a Leica EM Grid Plunger. Cryo-FIB milling was performed using either a Scios or Aquilos dual-beam FIB/SEM (Thermo Fisher Scientific). Grids were coated with a protective organometallic platinum layer, followed by trench milling for stress relief [64]. Final lamellas (~70–200 nm thick) were milled with a gallium ion beam to expose internal cellular structures. Lamellas were imaged on a Titan Krios TEM (300 kV, Thermo Fisher Scientific) equipped with a Gatan energy filter and a K2 Summit direct electron detector. Tilt series were acquired using SerialEM from −60° to +60° in 2° increments, employing a dose-symmetric tilt scheme [65]. Data were recorded in movie mode (12 fps) at a pixel size of 3.52 Å (~42,000× magnification), with a defocus range of −4 to −6 μm. The total electron dose per tilt series was ~100 e⁻/Å².

Tilt-series alignment and tomogram reconstruction were performed using TOMOMAN [66]. Denoising was applied using cryo-CARE [67] and IsoNet [68]. Subtomogram averaging and classification were carried out in STOPGAP [69].

## Cell cycle analysis

DNA synthesis during S-phase was assessed by 5-ethynyl-2′-deoxyuridine (EdU) incorporation. U2OS cells were seeded on 12 mm glass coverslips and transfected with either siCTRL or siC2CD3 as described above. Cells were incubated with

10 μM EdU for 1 h at 37 °C to allow incorporation into newly synthesized DNA. Following labeling, cells were fixed in 4% paraformaldehyde in PBS for 15 min at room temperature and processed for EdU detection using the Click-iT reaction according to the manufacturer's instructions (Thermo Fisher Scientific). After the Click reaction, cells were washed three times with PBS (10 min each) and counterstained with DAPI for 5 min to visualize nuclei, followed by three additional PBS washes. Coverslips were mounted in a glycerol-based mounting medium containing DAPI and DABCO (1,4-diazabicyclo[2.2.2]octane; Abcam, ab188804). Images were acquired using a Leica Thunder microscope and processed with the associated Leica software.

## Supporting information

**S1 Fig. C2CD3 displays a 9-fold radial localization at the distal end of centrioles. (A, B)** Expanded U2OS centrioles using iU-ExM and stained for tubulin in magenta. Nterm C2CD3 (A) or Middle C2CD3 (B) in green. Scale bar: 100 nm corrected by the expansion factor. **(C)** Additional examples of symmetrization of top-view images with the localization Nter C2CD3 in iU-ExM. Scale bar: 100 nm corrected by the expansion factor. **(D)** Corresponding intensity plot profiles highlighting the position of C2CD3 N-ter in between microtubule triplets. The detailed statistics of all the graphs shown in the figure are included in the S1 Data file.
(TIFF)

**S2 Fig. Visualization of the distal ring structure in situ in mouse retina photoreceptors.** Cryo-tomogram sections of an entire centriole from photoreceptor cells obtained by in situ cryo-electron tomography, shown from the distal (section 1) to proximal end (section 11). Each section represents an average of 31 slices (~30 nm total). A prominent ring-like structure is visible 30–60 nm from the distal tip.
(TIFF)

**S3 Fig. Structural organization of distal ring and hook elements in mouse tracheal epithelial cells. (A, B)** Nine-fold symmetrized cryo-tomogram slices of a centriole from mouse tracheal epithelial cells (MTECs). (A) The luminal ring structure is not visible, while the microtubule-associated "hook" structures are observed (pink arrowheads). (B) Both the luminal ring (orange arrowhead) and the hook structures (pink arrowheads) are visible and positioned within the same z-plane, indicating their coplanarity. White arrowheads mark microtubule triplets or doublets [67]. **(C)** Relative position of the hook structures with respect to the luminal ring. The graph shows the measured distances along the microtubule from the ring structure to the distal end of the hook structures, with an average distance of 28.96 nm ± 7.58 nm. **(D)** Fourier Shell Correlation (FSC) resolution estimation of the subtomograms average centered on the hook density between microtubule doublets/triplets. Resolution: 36 Å at an FSC cutoff of 0.143. The 3D reconstruction (right) shows continuous hook density along the A-microtubule wall (pink arrowheads). **(E)** FSC resolution estimation of the subtomograms average centered on the ring structure, containing three node densities. Resolution: 38 Å at an FSC cutoff of 0.143. The corresponding 3D map (right) highlights the amorphous density (blue arrowheads) and the structured ring elements (orange arrowheads). The detailed statistics of all the graphs shown in the figure are included in the S1 Data file.
(TIFF)

**S4 Fig. Molecular organization of the centriole distal end. (A)** Expanded centrioles from U2OS stained for α/β-tubulin (magenta) and NA14 (green) showing the localization of NA14 during centriole assembly. Scale bars: 100 nm. **(B)** Quantification of NA14 positioning relative to the distal end during assembly and in mature centrioles. **(C)** Average position of NA14 in mature centriole according to the tubulin proximal part of the mature centriole. Tubulin length: 438 ± 42; NA14 position: 408 ± 40 nm. **(D, E)** Distal end diameter measurements from side (D) and top (E) views. Tubulin: 139.21 ± 15.52 nm ($n = 43$, side), 179.57 ± 18.23 nm ($n = 15$, top); NA14: 70.85 ± 21.26 nm ($n = 43$, side), 80.03 ± 13.93 nm ($n = 15$, top). Data from three independent experiments. **(F)** Expanded centrioles from U2OS stained for α/β-tubulin

(magenta) and NA14 (green) showing the localization of NA14 in top views in procentriole and in mature centrioles. Scale bars: 100 nm. **(G)** Consecutive z-stack slices of expanded U2OS centrioles stained for α/β-tubulin (magenta) and NA14 (green), highlighting the distal localization of NA14 in top views. Scale bars: 200 nm. **(H, I)** Expanded U2OS centrioles using iU-ExM, stained for tubulin (magenta) and either in green NA14 (H) or SFI1 (I). Scale bar: 50 nm, corrected for expansion factor. **(J)** Dual localizations of Mid-C2CD3 with CEP90 and MNR, and NA14 with MNR and CEP90. Proteins of interest in green; tubulin in magenta. **(K)** Models of dual localization of Mid-C2CD3 with either CEP90 on the A-Microtubule (left), B-microtubule (middle) or C-microtubule (right). For each model, the angle formed by the CEP90 signal relative to the centriole center and the C2CD3 axis is measured and indicated in the figure. The detailed statistics of all the graphs shown in the figure are included in the S1 Data file.
(TIFF)

**S5 Fig. Overexpression of OFD1, CEP90, C2CD3, and MNR.** U2OS cells are transfected with **(A)** mCherry-OFD1, **(B)** CEP90, **(C)** C2CD3-GFP and CEP90, **(D)** C2CD3-GFP, and mCherry-OFD1, **(E)** MNR-mCherry and mCherry-OFD1, **(F)** MNR-mCherry and CEP90. When co-transfected with either CEP90 or OFD1, C2CD3 colocalizes in cytoplasmic granules. Both CEP90 and OFD1 are relocalized on microtubules when co-expressed with MNR. Percentage of GFP/mCherry positive cells with proteins localized to cytosol (Cyt.) or microtubules (MT) for each condition. (A) Cyt.: 100; MT: 0, (B) Cyt.: 0; MT: 100, (C) Cyt.: 100; MT: 0, (D) Cyt.: 97.62 ± 4.124; MT: 2.381 ± 4.124, (E) Cyt.: 1.515 ± 2.624; MT: 98.48 ± 2.624, (F) Cyt.: 0; MT: 100. Scale bar: 10 μm Three independent experiments performed for each condition. The detailed statistics of all the graphs shown in the figure are included in the S1 Data file.
(TIFF)

**S6 Fig. Impact of C2CD3 depletion on cell cycle progression and procentriole elongation. (A, B)** Representative fields of U2OS cells treated with siCTRL (A) or siC2CD3 (B), stained with DAPI (cyan) to label nuclei and EdU (yellow) to mark S-phase cells. **(C)** Percentage of cells in S phase as determined by Click-iT EdU labeling in siCTRL (dark gray) or siC2CD3 (light gray) conditions. siCTRL: 38.32% ± 6.97; siC2CD3: 32.08% ± 5.88. Statistical significance was determined using an unpaired $t$ test (*$P = 0.0150$). **(D)** Total number of cells counted per condition (ClickEdU). siCTRL: 392.1 ± 97.49; siC2CD3: 275 ± 50.50. Statistical significance was determined using an unpaired $t$ test (***$P = 0.0004$). **(E)** Representative images of expanded centrosomes in siCTRL and siC2CD3-treated cells, stained for tubulin (magenta) and SAS-6 (green). White arrows indicate the microtubule walls of procentrioles, which appear shorter in siC2CD3-treated cells compared to controls, despite having similar cartwheel (SAS-6) lengths. The detailed statistics of all the graphs shown in the figure are included in the S1 Data file.
(TIFF)

**S1 Data. Detailed statistics of all the graphs shown in the main and supporting information figures.**
(XLSX)

## Acknowledgments

We thank Jürgen Plitzko and Wolfgang Baumeister for access to cryo-FIB and cryo-EM instrumentation as well as the Dubochet Center for Imaging (DCI Lausanne and Geneva).

## Author contributions

**Conceptualization:** Eloïse Bertiaux, Tim Stearns, Benjamin D. Engel, Virginie Hamel, Paul Guichard.

**Data curation:** Eloïse Bertiaux, Vincent Louvel, Caitlyn L. McCafferty, Hugo van den Hoek, Umut Batman, Souradip Mukherjee, Lorène Bournonville, Olivier Mercey, Isabelle Méan, Jean Daraspe, Benjamin D. Engel.

**Formal analysis:** Eloïse Bertiaux, Vincent Louvel, Caitlyn L. McCafferty, Hugo van den Hoek, Umut Batman, Souradip Mukherjee, Lorène Bournonville, Olivier Mercey, Ricardo D. Righetto, Adrian Müller, Philippe Van der Stappen, Garrison Buss, Virginie Hamel, Paul Guichard.

**Funding acquisition:** Eloïse Bertiaux, Caitlyn L. McCafferty, Tim Stearns, Benjamin D. Engel, Virginie Hamel, Paul Guichard.

**Investigation:** Eloïse Bertiaux, Hugo van den Hoek, Umut Batman, Souradip Mukherjee, Jean Daraspe, Tim Stearns, Benjamin D. Engel, Virginie Hamel, Paul Guichard.

**Methodology:** Eloïse Bertiaux, Vincent Louvel, Hugo van den Hoek, Umut Batman, Souradip Mukherjee, Garrison Buss, Benjamin D. Engel, Virginie Hamel, Paul Guichard.

**Project administration:** Benjamin D. Engel, Virginie Hamel, Paul Guichard.

**Resources:** Christel Genoud, Virginie Hamel.

**Supervision:** Benjamin D. Engel, Virginie Hamel, Paul Guichard.

**Validation:** Eloïse Bertiaux, Hugo van den Hoek, Benjamin D. Engel, Virginie Hamel, Paul Guichard.

**Visualization:** Vincent Louvel, Caitlyn L. McCafferty, Hugo van den Hoek, Benjamin D. Engel, Paul Guichard.

**Writing – original draft:** Benjamin D. Engel, Virginie Hamel, Paul Guichard.

**Writing – review & editing:** Eloïse Bertiaux, Vincent Louvel, Caitlyn L. McCafferty, Umut Batman, Olivier Mercey, Tim Stearns, Benjamin D. Engel, Virginie Hamel, Paul Guichard.

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
