## [Editor Report · Decision Letter 0]

24 Jun 2025

Dear Dr Guichard,

Thank you for submitting your manuscript entitled "The Luminal Ring Protein C2CD3 Acts as a Radial In-to-Out Organizer of the Distal Centriole and Appendages" for consideration as a Research Article by PLOS Biology.

Your manuscript has now been evaluated by the PLOS Biology editorial staff as well as by an academic editor with relevant expertise and I am writing to let you know that we would like to send your submission out for external peer review.

Once your full submission is complete, your paper will undergo a series of checks in preparation for peer review. After your manuscript has passed the checks it will be sent out for review. To provide the metadata for your submission, please Login to Editorial Manager (https://www.editorialmanager.com/pbiology) within two working days, i.e. by Jun 26 2025 11:59PM.

Kind regards,

Ines

--

Ines Alvarez-Garcia, PhD

Senior Editor

PLOS Biology

---

## [Decision Letter · Decision Letter 1]

14 Aug 2025

Dear Paul,

Thank you for your patience while your manuscript "The Luminal Ring Protein C2CD3 Acts as a Radial In-to-Out Organizer of the Distal Centriole and Appendages" went through peer-review at PLOS Biology. Please note that I am currently handling your manuscript since my colleague Ines Alvarez-Garcia is away from the office this week on holiday. I am sorry for the delays that you have experienced during the peer review process. Your manuscript has now been evaluated by the PLOS Biology editors, an Academic Editor with relevant expertise, and by two independent reviewers.

As you will see, both reviewers are very positive about the manuscript and think it provides a significant advance in our understanding of the structural organization of centrioles. In light of the reviews, which you will find at the end of this email, we are pleased to offer you the opportunity to address the comments from the reviewers in a revision that we anticipate should not take you very long. We will then assess your revised manuscript and your response to the reviewers' comments with our Academic Editor aiming to avoid further rounds of peer-review, although we might need to consult with the reviewers, depending on the nature of the revisions.

IMPORTANT

In addition, I would be grateful if you could please address the following data-related requests and editorial requests that I have provided below (A-G):

(A) In the Methods section of the manuscript, please include the full name of the IACUC/ethics committee that reviewed and approved the mouse studies conducted in the paper. Please also include the specific approval number provided by the IACUC/animal ethics committee.

(B) You may be aware of the PLOS Data Policy, which requires that all data be made available without restriction: http://journals.plos.org/plosbiology/s/data-availability. For more information, please also see this editorial: http://dx.doi.org/10.1371/journal.pbio.1001797

-Supplementary files (e.g., excel). Please ensure that all data files are uploaded as 'Supporting Information' and are invariably referred to (in the manuscript, figure legends, and the Description field when uploading your files) using the following format verbatim: S1 Data, S2 Data, etc. Multiple panels of a single or even several figures can be included as multiple sheets in one excel file that is saved using exactly the following convention: S1_Data.xlsx (using an underscore).

-Deposition in a publicly available repository. Please also provide the accession code or a reviewer link so that we may view your data before publication.

Figure 1E-F, 2H, 3B-C, 3F, 4A-Q, 5C-L, S1D, S3C, S4B-E, S5, S6C-F

(C) Please deposit the cryo-ET data in a public data repository such as EMPIAR or EMDB. Please ensure that the data is made publicly available and provide the accession number in the Data Availability Statement in the online submission form.

(D) Please also ensure that each of the relevant figure legends in your manuscript include information on *WHERE THE UNDERLYING DATA CAN BE FOUND*, and ensure your supplemental data file/s has a legend.

(E) Please ensure that your Data Statement in the submission system accurately describes where your data can be found and is in final format, as it will be published as written there.

(F) Per journal policy, if you have generated any custom code during the course of this investigation, please make it available without restrictions. Please ensure that the code is sufficiently well documented and reusable, and that your Data Statement in the Editorial Manager submission system accurately describes where your code can be found.

(G) Please note that per journal policy, the model system/species studied should be clearly stated in the abstract of your manuscript.

**IMPORTANT - SUBMITTING YOUR REVISION**

*Resubmission Checklist*

*Published Peer Review*

*PLOS Data Policy*

*Blot and Gel Data Policy*

Best regards,

Richard

Richard Hodge, PhD

rhodge@plos.org

On behalf of:

Ines Alvarez-Garcia, PhD

REVIEWS:

Reviewer #1: Centrioles are nine-fold microtubule-based structures playing an important role in cell division, cellular polarity and as basal bodies, represent sort of ciliary anchors. Appendages important for centriole docking to the plasma membrane and ciliogenesis are present at the distal ends. As stated by the authors, C2CD2 has been suggested to play a role for recruitment of protein complexes playing an important role for appendage assembly and dysfunction of C2CD3 results in a ciliopathy phenotype. It has also been previously shown that C2CD3 appears as a ring-like structure forming the luminal distal ring (LDR) complex together with other proteins and that C2CD3 proteins are arranged in a 9-fold symmetrical arrangement positioned near the A-tubule. This suggested a scaffolding role for C2CD3 however the precise architectural mechanism by which C2CD3 establishes the nine-fold symmetry of distal appendage assembly has remained unclear.

In this paper, the authors investigate the precise molecular organisation of the distal centriolar region using advanced imaging techniques including U-ExM and cryo-electron tomography. They confirm the previously proposed nine-fold symmetry and suggest that the C2CD3 C-terminus localizes to the lumen while the C2CD3 N-terminus locates between microtubule blades with a"hook" connector between microtubule blades near the C2CD3 N-terminus. This explains previously reported discrepancy of diameters of 80-90 nm found using U-ExM, and a 146.4 nm ring with nine distinct puncta using U-ExM combined with STORM as the smaller diameter was seen using an antibody detecting the C-terminus of C2CD3 whereas the larger diameter was seen using an antibody detecting the middle part of C2CD3. The authors map out precise positions of C2CD3 and other proteins part of the appendage protein complexes, C2CD3 might bridge the centriole lumen to the external DISCO complex, forming an in-to-out connection through the space between microtubule triplets/doublets.

Using displacement assays using overexpression, they show that C2CD3 does not bind microtubules when overexpressed alone but when co-expressed with MNR and that overexpression of MNR with C2CD3, C2CD3/CEP90, and C2CD3/OFD1/CEP90 led to increased microtubule localization, suggesting a cooperativity in the binding. They further show C2CD3 interacts with appendage-anchoring proteins MNR and CEP90, and contributes to a broader distal complex including SFI1, Centrin-2, CEP135, and NA14/ SSNA1 with NA14/ SSNA1 co-localizing with the C2CD3 C-terminus.

They also found that depletion of C2CD3 using siRNA results in shorter centrioles with abnormal CEP97 and CEP162 localization while core structural features of the proximal and middle centriole regions are intact. From this they conclude C2CD3 plays an important role for distal centriole architecture as well as microtubule cap composition.

Overall, this is a very well written manuscript with superb imaging data and 3D reconstructions, providing novel insights into the structural organisation of centrioles and the precise role of C2CD3. I feel this will be interesting to a broad readership and only have very few comments.

1. siRNA knockdown: I couldn't find data on protein expression in control vs siRNA cells. This should be added to the supplement.

2. Figure S6 C suggest a statistically significant reduction of cells in S Phase (something like 38% vs 32%) in C2CD3 siRNA cells vs siRNA controls while in figure S6 E and F, bars suggest that in both groups, less than 20% were in S Phase with no striking difference. Does this difference between C and E/F stem from different methods used and is there a statistically significant difference between the two groups in E and F, also with regards to other cell cycle phases?

Reviewer #2 (Maxence Nachury, identifies himself): This manuscript presents an unparalleled combination of iterative expansion microscopy and cryo-electron tomography to paint the spatial organization of C2CD3 from inside the centriole to the inter-MT space of the centriolar barrel.

The manuscript explores the distal luminal ring complex, a structure first visualized in the 1970s, but with no molecular characterization to speak of. By carefully matching the locations of C2CD3-Cterm determined by U-ExM and those of the distal luminal ring complex mapped by cryoET, the authors propose that C2CD3 forms part of the distal luminal ring complex.

A major advance of the manuscript is the modeling of the distal ring structure from cryoET data, which consists of a 100nm ring with 27 nodes connected to nine-fold symmetrical amorphous linkers and hook anchored onto the A microtubule.

The rigor, precision and breadth of the work allows the authors to assemble a persuasive model for C2CD3 spanning the distal luminal ring to the DISCO complex on the outside of the centriole by fitting through the microtubule triplet blades.

Adding to the novelty, the authors identify the protein NA14 as part of the distal luminal ring.

The graphical summaries are very helpful to orient the non-specialist through the protein landscape of the distal centriole.

I have no major issue with the manuscript and strongly recommend publication.

Minor: scale bars are missing from several panels (1L, 3E, leftmost panel in 3A, bottom panels in 3D, 4D/E/F/H/I/J/L/M/N/P/Q, 5G-L, S1A-C, S4A, S4F)

I was confused about the graph in 3J. Shouldn't the bar be colored yellow if they depict C2CD3 distribution?

---

## [Editor Report · Decision Letter 2]

9 Nov 2025

Dear Dr Guichard,

Thank you for the submission of your revised Research Article entitled "The Luminal Ring Protein C2CD3 Acts as a Radial In-to-Out Organizer of the Distal Centriole and Appendages" for publication in PLOS Biology. On behalf of my colleagues and the Academic Editor, Dagmar Wachten, I am delighted to let you know that we can in principle accept your manuscript for publication, provided you address any remaining formatting and reporting issues. These will be detailed in an email you should receive within 2-3 business days from our colleagues in the journal operations team; no action is required from you until then. Please note that we will not be able to formally accept your manuscript and schedule it for publication until you have completed any requested changes.

PRESS

Sincerely, 

Ines

--

Ines Alvarez-Garcia, PhD

Senior Editor

PLOS Biology
